



# Validation of 12 years (2008-2019) of IASI-CO with IAGOS aircraft observations

Brice Barret[1], Pierre Loicq[1], Eric Le Flochmoën[1], Yasmine Bennouna[1], Juliette Hadji-Lazaro[2], Daniel Hurtmans[3], and Bastien Sauvage[1]

[1]LAERO/OMP, Université de Toulouse III - Paul Sabatier, CNRS, France
[2]LATMOS/IPSL, Sorbonne Université, UVSQ, CNRS, Paris, France
[3]Spectroscopie de l'Atmosphère, Chimie Quantique et Photophysique, Université Libre de Bruxelles, Belgium

**Correspondence:** B. Barret
(brice.barret@aero.obs-mip.fr)

**Abstract.**

    IASI-A, B and C (Infrared Atmospheric Sounding Interferometer) are nadir looking thermal infrared sensors which are monitoring the atmospheric composition since 2008. Atmospheric carbon monoxide (CO) is retrieved from IASI radiances with two algorithms: the SOftware for a Fast Retrieval of IASI Data (SOFRID) and the Fast Optimal Retrievals on Layers for IASI

(FORLI). The airborne in-situ observations from the In-service Aircraft for a Global Observing System (IAGOS) European Research Infrastructure have been used to validate the IASI CO retrievals. The validation study of IASI CO data performed in 2011 whith IAGOS data was limited to two airports (Frankfurt and Windhoek) and 2 years because of the limited sampling at the other IAGOS sites. The extension of the IAGOS infrastructure during the last decade enables a validation with enough temporal sampling at 33 airports worldwide over the whole IASI-A period (2008-2020).

The retrievals provide between 1.5 and 3 independent pieces of information about the CO vertical profile and we have selected to validate the surface-600 hPa and 600-200 hPa partial columns in addition to the total column. The ability of the retrievals to capture the CO variabilities is slightly different for the two retrieval algorithms. The correlation coefficients are generally larger for SOFRID, especially for the total and lower tropospheric columns, meaning a better representation of the phase of the variability, while the amplitude of the variations of FORLI are in better agreement with IAGOS in the mid-upper troposphere.

On average SOFRID and FORLI retrievals are underestimating the IAGOS total columns of CO (TCC) by 8±16% and 6±14% respectively. This global TCC agreement between the algorithms is hiding significant vertical and geographical differences. In the lower troposphere (Surface-600 hPa) the bias is larger for FORLI (-11±27%) than for SOFRID (-4±24%). In the mid-upper troposphere the situation is reversed with a bias of -6±15% for FORLI and of -11±13% for SOFRID. The largest differences between the retrievals are detected south of Bangkok where SOFRID underestimation is systematically larger for the TCC and

mid-upper tropospheric column. North of Philadelphia FORLI biases are significantly larger than SOFRID ones for the TCC and the lower tropospheric columns. Our validation results will provide a better characterisation of IASI-CO data to the users and help improve the retrievals for future versions.



## 1 Introduction

The largest sources of carbon monoxide (CO) in the atmopshere are biomass burning and fossil fuel combustion from anthropogenic activities. The oxidation of methane ($CH_4$) and non-methane hydrocarbons still accounts for the production of about half of the CO global burden. The main sink of CO ($\sim 90\%$) is its oxidation by the hydroxyl radical (OH) (Lelieveld et al., 2016). CO is thereby impacting the oxidizing capacity of the atmosphere and the lifetime of $CH_4$ (Bergamaschi et al.) which is the second most important greenhouse gas of anthropogenic origin. Through its oxidation in the presence of nitrogen oxides

(NOx), CO is also involved in the production of tropospheric $O_3$. Finally, its lifetime in the troposphere of 1 to 2 months makes of CO a good tracer of pollution long-range transport (Forster et al., 2001).

The IASI (Infrared Atmospheric Sounding Interferometer) sensors launched onboard MetOp-A (2006), B (2012) and C (2018) allow the monitoring of meteorological parameters (water vapour and temperature) and of a number of atmospheric trace species with an unprecendented spatio-temporal coverage (Clerbaux et al., 2009). Two algorithms have been developed for

the retrieval of vertical profiles of CO from IASI: the SOftware for a Fast Retrieval of IASI Data (SOFRID) (De Wachter et al., 2012) and the Fast Optimal Retrievals on Layers for IASI (FORLI, (Hurtmans et al., 2012; George et al., 2009). These retrievals have been used intensively to document biomass burning (Bencherif et al., 2020; Turquety et al., 2020), urban pollution (Stremme et al., 2013; Yarragunta et al., 2019), long-range transport and convection uplift of pollution (Lannuque et al., 2021; Barret et al., 2016; Tsivlidou et al., 2022), and the COVID-19 lockdowns impact on air quality (Zhou et al., 2021;

Clark et al., 2021). FORLI CO data have been compared with data from the Measurement of Pollution in the Troposphere (MOPITT) highlighting the significant impact of the a priori information on the retrieval differences (George et al., 2015). In Buchholz et al. (2021), decadal CO trends were estimated from long-term MOPITT data and FORLI retrievals displayed consistent hemispheric CO variability and corroborated the results. FORLI-CO total columns from IASI/Metop-C have been recently validated with NDACC-FTIR data (https://acsaf.org/docs/vr/Validation_Report_IASI-C_CO_May_2021.pdf). SOFRID

and FORLI retrieved profiles from IASI-A have been validated against airborne in-situ data from the In-service Aircraft for a Global Observing System (IAGOS) European Research Infrastructure for years 2008 and 2009 at the airports of Frankfurt in Germany and Windhoek in Namibia in De Wachter et al. (2012). Since that time, the IASI retrievals have evolved with a number of successive versions and the IAGOS infrastructure has been extended to many airports, particularly in Asia and we benefit from longer time series. The purpose of this paper is therefore to validate the 12 years of IASI-A CO retrievals

with the extended IAGOS database in order to (i) have a validation covering a large number of regions especially Asia where anthropogenic pollution is the most important and (ii) document the time stability of the retrievals focusing on sites providing dense and continuous time series such as Frankfurt.

The paper is structured as follows. We start with the presentation of the IASI retrievals and the IAGOS data in section 2. The methodology of validation is introduced in section 3.2 and the results are presented under three different aspects: the compar-

ison of variabilities (section 3.3.1), the biases (section 3.3.2) and the time series with the temporal variabilities at the airports with the densest and longest IAGOS datasets (section section 3.4). The synthesis of the main results are finally provided in the





conclusions.

## 2 Data

### 2.1 SOFRID-CO IASI retrievals

SOFRID-CO allows the fast retrieval of CO profiles on 43 levels from the ground up to 0.1 hPa from MetOp/IASI radiance measurements (De Wachter et al., 2012). It is based on the RTTOV (Saunders et al., 1999; Matricardi et al., 2004; Matricardi and al., 2009) fast radiative transfer model coupled to the UKMO 1D-Var retrieval scheme (Pavelin et al., 2008) based on the optimal estimation method (OEM) described by Rodgers (2000). In the present study we use SOFRID-CO v4.0 which has

been updated since the v2.0 used in De Wachter et al. (2012). First, SOFRID-$N_2O$ (Barret et al., 2021) was recently developed to retrieve the $N_2O$ profiles from a spectral window (2160-2218 $cm^{-1}$) partly overlapping the CO window (2143-2181 $cm^{-1}$) from De Wachter et al. (2012). In order to retrieve $N_2O$ together with CO we have merged the retrieval windows of CO and nitrous oxide ($N_2O$) to 2143-2218 $cm^{-1}$. The meteorological parameters needed for the radiative transfer calculations (surface pressure, temperature and humidity profiles) are taken from ECMWF operational analyses instead of operational EUMETSAT

Level 2 IASI products. RTTOV has been updated from v9.3 to v12.3 and we use the UKMO 1D-Var v1.2. The noise of the measurement covariance matrix has been reduced from 1.4 to $1.0 \cdot 10^{-8}$ W/($cm^2$ sr $cm^{-1}$). The a priori covariance matrix is the one from De Wachter et al. (2012) for CO and $H_2O$ and from Barret et al. (2021) for $N_2O$. We only retrieve CO from pixels with a cloud fraction less than 25% as in De Wachter et al. (2012). SOFRID CO daily and monthly data are available for the whole period through the Service de données de l'Observatoire Midi-Pyrénées (https://iasi-sofrid.sedoo.fr/).


### 2.2 FORLI-CO IASI retrievals

In FORLI, CO retrievals are performed in the 2143–2181.25$cm^{-1}$ spectral range chosen to minimize interferences by carbon dioxide, $N_2O$ and ozone, using the OEM and tabulated absorption cross sections at various pressures and temperatures to speed up the radiative transfer calculation. A priori information consists in one single CO a priori profile and one single

covariance matrix based on a set of model, satellite and aircraft profiles (Hurtmans et al., 2012). The EUMETSAT Level 2 data (pressure, water vapor, temperature and cloud information) used as input in FORLI have been processed using different versions of the IASI Level 2 Product Processing Facility between 2008 (v4.2) and 2016 (v6.2) (Van Damme et al., 2017). Retrievals are only processed for scenes with a fractional cloud cover from the EUMETSAT operational processing (August et al., 2012) below 25%. In addition, no retrieval is performed for pixels characterized by a Level 1C error or by

missing L2 EUMETSAT data. FORLI-CO provides vertical profiles in 18 layers between the surface and 18 km, with an extra layer from 18 km to the top of the atmosphere. FORLI-CO data also include a general quality flag, the total error profile and the averaging kernel (AK) matrix. For this validation study FORLI-CO v20151001 was used. This version was installed




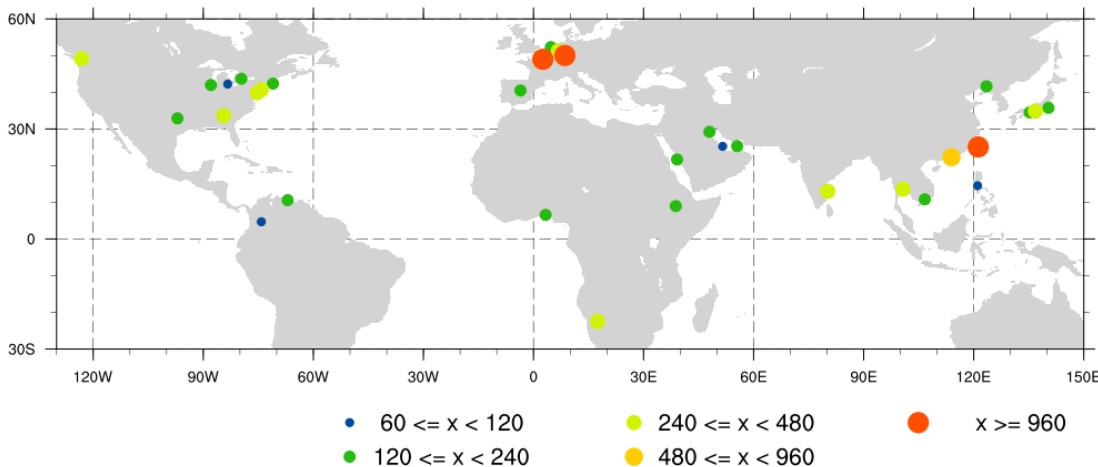

**Figure 1.** Position of the 33 selected airports with IAGOS data for the 2008-2020 period. The size of the symbols is proportional to the number of valid days used for validation.

in EUMETSAT Ground Segment in the AC SAF (https://www.eumetsat.int/ac-saf) framework to generate the CO product (https://navigator.eumetsat.int/product/EO:EUM:DAT:METOP:IASIL2COX).


### 2.3 IAGOS airborne in-situ data

We use CO in-situ observations from the IAGOS European Research Infrastructure (Nedelec et al. (2015); Petzold et al. (2015), https://www.iagos.fr). CO is measured using a dual-beam ultraviolet absorption monitor (infrared analyser) with an accuracy of 5 ppbv, a precision of 5% and a time resolution of 30 seconds (Blot et al., 2021). Vertical profiles are recorded during ascend

and descend phases. Considering the aircraft vertical speed (7-8 m.s$^{-1}$), the vertical resolution is about 450 m. CO observations are collected since 2002 based on the same technology.

From the IAGOS database, only airports providing at least 60 days with valid data between 2008 and 2020 were selected. This selection criterion leads to 33 airports representing a total of 14211 profiles (8478 days). The locations of these 33 airports are given in Table A1 and displayed on Fig. 1. The temporal availabilities of the IAGOS data are also displayed for each of the 33

airports on Fig. 2.

Frankfurt represents 35% (4917 profiles) of these observations and provides the longest and most continuous time serie. In Europe the other 4 airports have much less observations over shorter time periods. Over Northern America (9 airports), Atlanta represents the longest and densest time serie covering the full period with some important gaps. Time series over Asia (10 airports) are mostly starting after 2012 except for Nagoya starting in late 2009. Taipei and Bangkok provide dense and long

time series for South-East and East Asia. During the IASI-A period, only 3 airports have been sampled by IAGOS over Africa





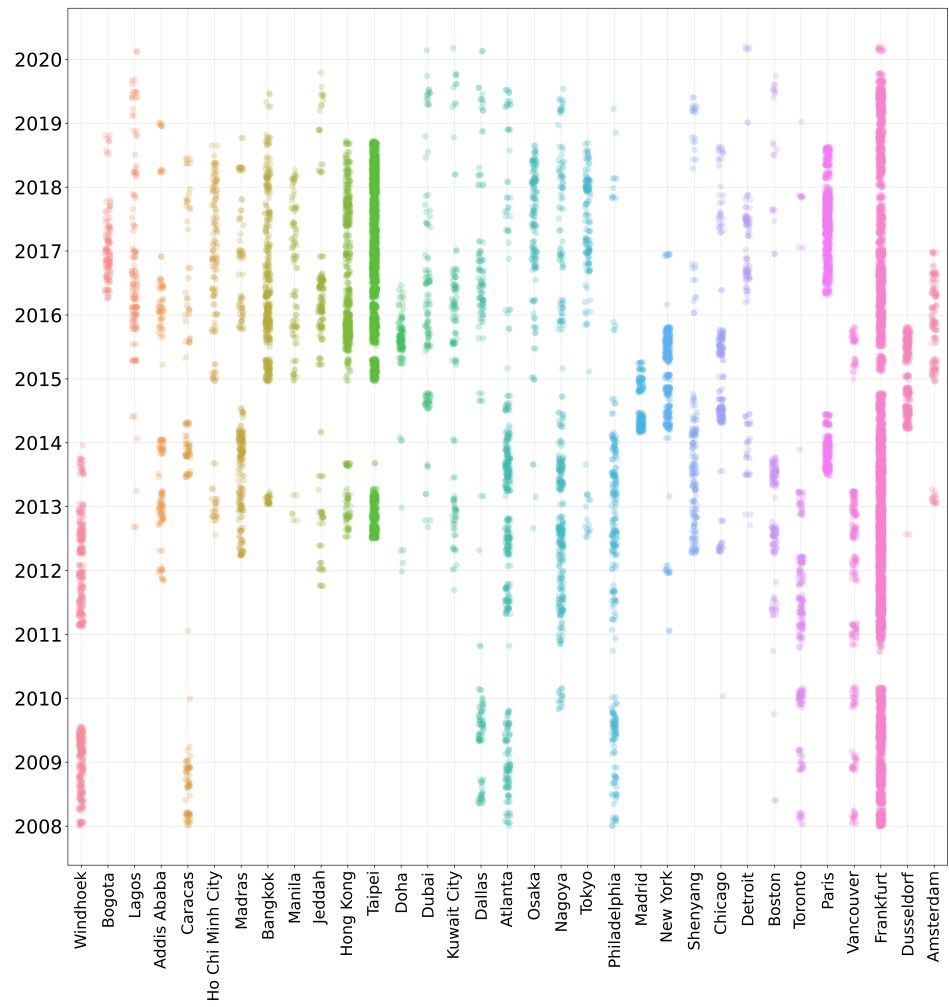

**Figure 2.** Availability of IAGOS profiles at the 33 airports for the 2008-2020 period.

among which Windhoek provides the longest and densest time serie. The four Middle East airports are mostly covering the period 2015-2016 with some sparse data earlier and later. Over South America, Caracas provides sparse data from 2008 to 2018 and Bogota less than 3 years with data. In section 3.4 timeseries are shown for the airports with the densest and longest IAGOS datasets: Frankfurt, Atlanta, Bangkok, Taipei, Nagoya, and Windhoek.




| Layer | FORLI DFS | SOFRID DFS |
|---|---|---|
| TCC | 1.6 | 2.9 |
| Surface-600 hPa | 0.6 | 1.1 |
| 600-200 hPa | 0.9 | 1.3 |

**Table 1.** DFS for FORLI and SOFRID for total and partial columns averaged over the validation dataset at the 33 selected airports.

## 3 Validation

### 3.1 Information content analysis

The vertical sensitivity of the retrievals is characterised by the Averaging Kernels (AK) matrix. For each retrieval layer, the retrieved quantity is the result of the convolution of the real profile by the corresponding averaging kernel (row of the AK matrix)
plus a contribution from the a priori profile ($\mathbf{x}_a$) (see Eq. 1). The AKs are bell shaped functions of which the width gives an indication of the retrieval vertical resolution. The trace of the AK matrix called Degrees of Freedom for Signal (DFS) provides the number of independent pieces of information about the vertical profile from the retrieval. The AKs at Frankfurt averaged over the validation datasets are displayed in Fig. 3 for FORLI and SOFRID for the winter (December-January-February - DJF) and summer (June-July-August - JJA) seasons. The DFS for the total colums of CO (TCC) and two selected partial columns
averaged over the 33 airports (section 2.3) are given in Table 1. For FORLI the retrievals provide a total of 1.6 independent pieces of information against 2.9 for SOFRID. The larger information content from SOFRID is due to (i) the extension of the spectral window and (ii) the reduction of the noise of the measurement covariance matrix relative to De Wachter et al. (2012). Both modifications are related to the combination of CO with $N_2O$ retrievals.

For both algorithms, the DFS are larger at Frankfurt in JJA than in DJF (Fig. 3) because the surface temperature and the surface-atmosphere thermal contrast is larger in summer. The JJA individual AKs for FORLI display roughly two groups with one corresponding to layers between 900 and 700 hPa that peak in the lower troposphere and the second one corresponding to layers between 500 and 250 hPa which are sensitive to the mid- and upper troposphere. In DJF there is only one distinct group of AKs with maximum sensitivity between 700 and 200 hPa. For SOFRID and for both seasons, the AKs display roughly 3
groups with maximum sensitivity at about 900, 500 and 150 hPa. We have therefore selected the layers surface-600 hPa and 600-200 hPa as the two pieces of information that can be retrieved by both algorithms. The average DFS for these two partial columns range from 0.6 for FORLI in the lower layer in DJF to 1.3 for SOFRID in the upper layer in JJA which confirm that they correspond to almost independent pieces of information (Table 1). For the TCC, the retrieval errors (sum of the measurement and smoothing errors (Rodgers, 2000)) provided with the retrievals are similar for both algorithms with a mean value of
135 5%.



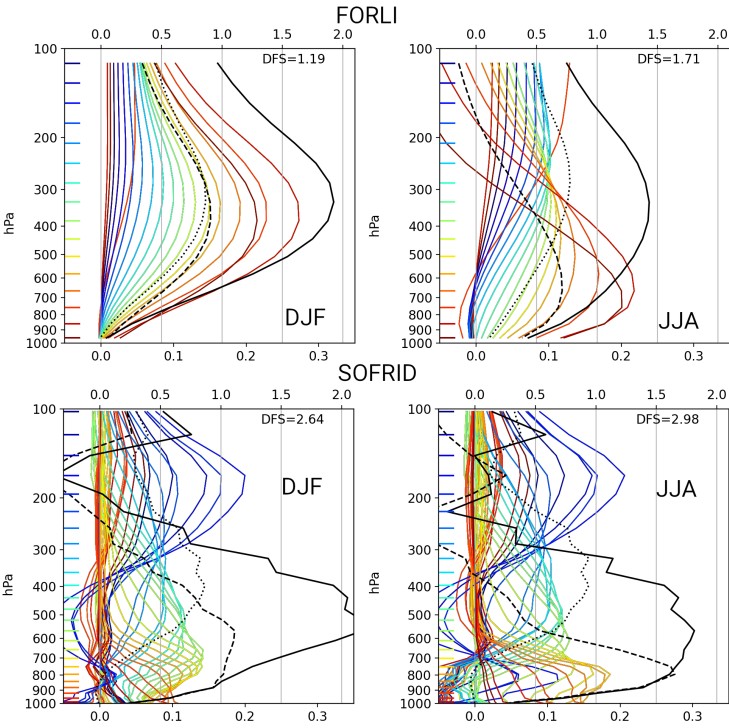

**Figure 3.** FORLI (top) and SOFRID (bottom) averaging kernels (bottom x-axis, color lines) and normalised averaging kernels for integrated columns (top x-axis, black solid line (total column), black dashed line (surface-600 hPa), black dotted line (600-200 hPa)), for daytime retrievals averaged over the validation database at Frankfurt for (left) DJF and (right) JJA. The nominal height of each averaging kernel is marked by the horizontal tick with the corresponding colour.

The AKs for the two partial columns have clearly distinct peaks for SOFRID and for FORLI in JJA (Fig. 3). In DJF, FORLI's AKs display a single peak following the low total information content (1.19). For the two other seasons (MAM and SON, not shown) the DFS for FORLI is about 1.5 and the AKs for the partial columns are similar to the AKs for JJA. It is noteworthy that for the different seasons and both algorithms the AKs display minima at the surface indicating a low sensitivity in the Boundary Layer (Fig. 3).

## 3.2 Methodology

The validation methodology is presented in the flow-chart of Fig. 4. The IAGOS profiles are selected according to their vertical completeness below the aircraft cruise altitude. Furthermore, they are completed in the upper troposphere and stratosphere with Aura MLS v5.0 CO profiles filtered according to data quality (Livesey et al., 2020) and averaged in 5° latitude x 5° longitude boxes over 5 days with a procedure similar to the one described in De Wachter et al. (2012). Both IAGOS and MLS profiles



are interpolated on the 19 FORLI and 43 SOFRID retrieval levels and merged.

IASI pixels were extracted in squares of $\pm 1°$ in latitude and longitude around the aircraft position at 6 km a.s.l on the same day as the corresponding take off or landing IAGOS profile (Fig. 4). We have chosen 6 km to be about half way between the ground and the cruise altitude. Pixels were filtered according to their retrieval quality. In order to take the retrieval vertical sensitivity and a priori impact into account for comparison, the IAGOS profiles, $\boldsymbol{x}_{IAG}$ were smoothed with the SOFRID and FORLI AK matrices (**A**) according to the following equation:


$$\hat{\boldsymbol{x}}_{IAG} = \boldsymbol{x}_a + \mathbf{A} \cdot (\boldsymbol{x}_{IAG} - \boldsymbol{x}_a) \tag{1}$$

where $\hat{\boldsymbol{x}}_{IAG}$ is the smoothed or convolved IAGOS profile and $\boldsymbol{x}_a$ is the a priori profile of the SOFRID or FORLI retrieval. The partial columns for the selected surface-600 hPa and 600-200 hPa layers and the TCC were computed for the IAGOS (raw and smoothed) profiles and for the SOFRID and FORLI retrievals. For each day and airport with a IAGOS profile, all

coinciding IASI and IAGOS (raw and smoothed) profiles were averaged.

### 3.3  General statistics

In this section, we present the comparisons of the results (TCC, surface-600 hPa, and 600-200 hPa) from SOFRID and FORLI with the data provided by IAGOS-MLS association, both raw and smoothed. The validation of satellite retrievals with indepen-

dent data requires to compute a number of indicators that quantify the ability of the retrievals to reproduce the absolute values and the variations of the retrieved quantity. The relative or absolute biases document the accuracy of the retrievals. The root mean squares of the differences (RMSD) between the two datasets inform about the significance of the biases. The Pearson (or correlation) coefficients (R) describe the agreement between the phases of the variabilities of the two datasets. Finally the ratios of the standard deviations document the agreement between the amplitudes of the variations.


### 3.3.1  Variabilities

The Taylor diagram used for climate model validation (Taylor, 2001) is taking advantage of the relationship between R, the RMSD and the variabilities (standard deviations) of two datasets to display synthetically these 3 parameters. Figure 5 presents Taylor diagrams comparing SOFRID and FORLI three columns with IAGOS raw data. RMSD between SOFRID/FORLI and

IAGOS datasets and standard deviations of SOFRID and FORLI results are normalised by the standard deviation of the reference, IAGOS raw data, to display the results from multiple experiments (here multiple airports) on a single diagram. We only display the Taylor diagrams for comparison of FORLI and SOFRID with raw IAGOS data because they provide the best assessment of the real differences between the in-situ and the remote sensed data. The Taylor diagrams for smoothed IAGOS



**Figure 4.** Flow chart of the validation methodology.





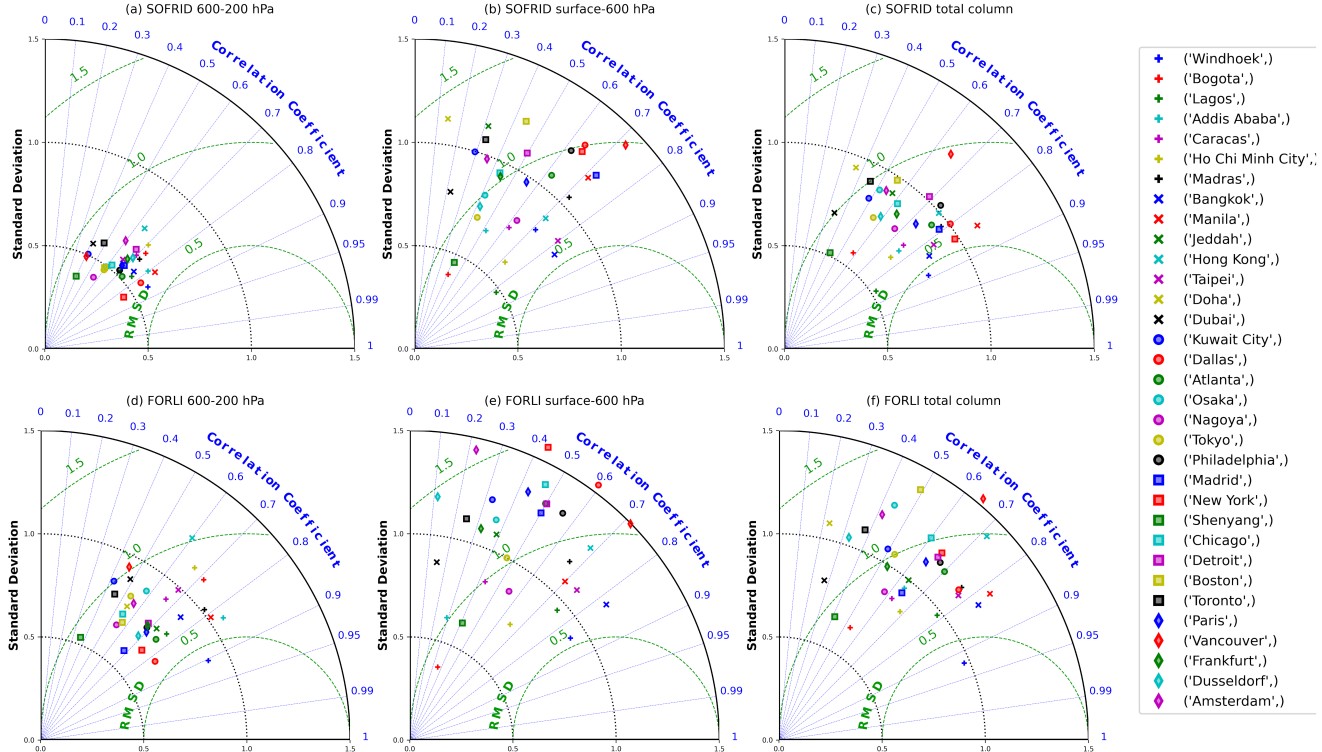

**Figure 5.** Taylor diagrams for the SOFRID (top) and FORLI (bottom) versus IAGOS raw data comparisons for the 600-200 hPa (left) Surface-600 hPa (middle) and Total CO columns (right).

data are provided in the Appendix (Fig. A1).


The reference (here IAGOS data) corresponds to graduation 1 on the X-axis (see Fig. 5). The RMSD is proportional to the distance from this reference point (green arcs of circle centered on reference point). The Pearson coefficient between the reference (IAGOS) and the test datasets (SOFRID and FORLI) is given by the azimuthal position of the point. Finally, the radial distance from the origin is proportional to the standard deviation of the experiment (i.e. retrievals of one of the algorithms at a

given airport). Each airport is represented by a marker of different shape and color. The better the agreement between SOFRID/-FORLI results and IAGOS raw data, the closer the markers will be to the reference point. For example, the point corresponding to Windhoek shows a better agreement for the TCC retrieved by FORLI while the agreement is better for SOFRID at New-York.

The biases, their standard deviations and the Pearson coefficients are also reported for TCC comparisons with raw and
smoothed IAGOS data in Table 2 (airports are listed in ascending order of latitude). For the two partial columns, the data are provided in Appendix A: Table A2 for surface-600 hPa and Table A3 for 600-200 hPa.



| Airport | FORLI | | | | SOFRID | | | |
|---|---|---|---|---|---|---|---|---|
| | Raw | IAGOS | Smoothed | IAGOS | Raw | IAGOS | Smoothed | IAGOS |
| | R | Bias % | R | Bias % | R | Bias % | R | Bias % |
| Windhoek | 0.92 | -4 ± 11 | 0.93 | -7 ± 10 | 0.89 | -11 ± 14 | 0.89 | -11 ± 14 |
| Bogota | 0.53 | -13 ± 16 | 0.73 | -9 ± 10 | 0.58 | -25 ± 13 | 0.63 | -23 ± 11 |
| Lagos | 0.78 | -10 ± 20 | 0.81 | -1 ± 19 | 0.85 | -20 ± 19 | 0.91 | -9 ± 13 |
| Addis Ababa | 0.64 | 0 ± 14 | 0.79 | -6 ± 10 | 0.76 | -15 ± 12 | 0.80 | -14 ± 10 |
| Caracas | 0.62 | -13 ± 13 | 0.67 | -11 ± 12 | 0.75 | -6 ± 10 | 0.76 | -5 ± 10 |
| Ho Chi Minh City | 0.69 | -18 ± 17 | 0.75 | -8 ± 15 | 0.76 | -20 ± 16 | 0.80 | -11 ± 13 |
| Madras | 0.77 | -2 ± 12 | 0.80 | -3 ± 12 | 0.79 | 0 ± 11 | 0.80 | 2 ± 10 |
| Bangkok | 0.83 | -2 ± 15 | 0.78 | 2 ± 18 | 0.84 | -10 ± 13 | 0.84 | -6 ± 13 |
| Manila | 0.82 | -1 ± 12 | 0.85 | 2 ± 12 | 0.84 | -2 ± 10 | 0.86 | 2 ± 10 |
| Jeddah | 0.63 | -6 ± 11 | 0.72 | -10 ± 12 | 0.57 | -2 ± 12 | 0.61 | -0 ± 11 |
| Hong Kong | 0.71 | -1 ± 20 | 0.70 | 1 ± 21 | 0.75 | -5 ± 15 | 0.76 | 0 ± 15 |
| Taipei | 0.78 | -4 ± 16 | 0.79 | -4 ± 15 | 0.82 | -9 ± 13 | 0.82 | -5 ± 14 |
| Doha | 0.23 | 1 ± 14 | 0.40 | 0 ± 14 | 0.37 | 3 ± 12 | 0.46 | 5 ± 11 |
| Dubai | 0.27 | -10 ± 13 | 0.48 | -6 ± 11 | 0.34 | -4 ± 12 | 0.41 | 2 ± 10 |
| Kuwait City | 0.49 | -8 ± 12 | 0.58 | -6 ± 12 | 0.49 | -3 ± 11 | 0.51 | 2 ± 10 |
| Dallas | 0.77 | -4 ± 11 | 0.78 | -6 ± 12 | 0.80 | -2 ± 9 | 0.80 | -1 ± 9 |
| Atlanta | 0.70 | -8 ± 11 | 0.71 | -7 ± 13 | 0.76 | -2 ± 9 | 0.77 | -1 ± 9 |
| Osaka | 0.44 | -0 ± 18 | 0.53 | -0 ± 17 | 0.51 | -5 ± 13 | 0.54 | -3 ± 14 |
| Nagoya | 0.58 | -8 ± 18 | 0.61 | -9 ± 19 | 0.68 | -7 ± 15 | 0.67 | -5 ± 16 |
| Tokyo | 0.53 | -5 ± 13 | 0.59 | -6 ± 14 | 0.56 | -6 ± 11 | 0.60 | -4 ± 10 |
| Philadelphia | 0.67 | -10 ± 12 | 0.72 | -10 ± 13 | 0.74 | -1 ± 10 | 0.74 | 0 ± 10 |
| Madrid | 0.64 | -12 ± 9 | 0.64 | -15 ± 10 | 0.79 | -10 ± 7 | 0.77 | -10 ± 8 |
| New York | 0.66 | -2 ± 13 | 0.68 | -4 ± 15 | 0.84 | 1 ± 8 | 0.82 | 2 ± 9 |
| Shenyang | 0.41 | -25 ± 28 | 0.44 | -18 ± 27 | 0.43 | -21 ± 27 | 0.43 | -14 ± 25 |
| Chicago | 0.60 | -5 ± 13 | 0.55 | -8 ± 15 | 0.61 | -3 ± 10 | 0.60 | -2 ± 12 |
| Detroit | 0.66 | -11 ± 11 | 0.69 | -11 ± 12 | 0.69 | -7 ± 9 | 0.70 | -5 ± 9 |
| Boston | 0.49 | -7 ± 16 | 0.45 | -10 ± 18 | 0.56 | 2 ± 12 | 0.55 | 2 ± 12 |
| Toronto | 0.38 | -20 ± 13 | 0.49 | -18 ± 16 | 0.46 | -8 ± 12 | 0.43 | -8 ± 12 |
| Paris | 0.64 | -6 ± 12 | 0.68 | -7 ± 12 | 0.72 | -1 ± 9 | 0.74 | -1 ± 8 |
| Vancouver | 0.65 | -14 ± 17 | 0.64 | -17 ± 17 | 0.65 | -12 ± 14 | 0.61 | -13 ± 15 |
| Frankfurt | 0.53 | -11 ± 14 | 0.66 | -11 ± 12 | 0.64 | -5 ± 11 | 0.65 | -5 ± 11 |
| Dusseldorf | 0.33 | -3 ± 16 | 0.41 | -4 ± 15 | 0.59 | 2 ± 11 | 0.62 | 4 ± 10 |
| Amsterdam | 0.42 | -8 ± 14 | 0.51 | -9 ± 13 | 0.54 | -4 ± 10 | 0.53 | -4 ± 10 |
| All | 0.78 | -8 ± 16 | 0.80 | -7 ± 16 | 0.81 | -6 ± 14 | 0.82 | -4 ± 13 |

**Table 2.** Pearson coefficients and biases for FORLI and SOFRID for total columns comparisons with raw and smoothed IAGOS data at the 33 selected airports listed in ascending order of latitude.





For IAGOS raw data and concerning the TCC, R is generally larger for SOFRID than for FORLI with for instance less points above the R=0.8 line (7 for SOFRID and 3 for FORLI) on the diagram (Fig. 5) or 24 airports with R<0.7 (r<0.5) for

FORLI against 17 for SOFRID (Table 2). The most striking example is at Dusseldorf where R=0.59 for SOFRID and 0.33 for FORLI (Table 2). On the contrary variabilities (standard deviations) are larger for FORLI than for SOFRID. For instance, at Lagos FORLI has the same amplitude of variations than IAGOS and SOFRID just half of it. At Vancouver FORLI's variations are about 1.5 times larger than IAGOS when SOFRID display an amplitude of variations closer to IAGOS. For 11 (resp. 14) airports the ratios of standard deviations between retrievals and IAGOS raw data are between 0.9 and 1.1 for FORLI (resp.

SOFRID). For 4 (resp. 12) airports these ratios are between 0.7 and 0.9 for FORLI (resp. SOFRID). Symmetrically, for 13 (resp. 3) airports they are ranging from 1.1 to 1.3 for FORLI (resp. SOFRID). For the remaining airports, SOFRID (resp. FORLI) underestimates (resp. overestimates) this amplitude. Therefore SOFRID reproduces slightly better the phase of the temporal variations of TCC while both algorithms capture the amplitude of these variations for about one third of the airports.

For the lower tropospheric column (surface-600 hPa) SOFRID and FORLI display larger spreads of indicators across the Taylor diagrams. For instance, RMSD ranges between 0.55 and 1.4 times IAGOS standard deviations for SOFRID and between 0.5 and 1.8 for FORLI (Fig. 5 (b) and (e)). For FORLI, Doha and Boston's variabilities are resp. 1.65 and 1.77 larger than IAGOS and the corresponding points are therefore out of the Taylor diagram. As for the TCC, SOFRID Pearson's coefficients are larger than FORLI's for a majority of airports (23, Table A2). For 5 airports SOFRID and FORLI's Pearson's

coefficients are equal or nearly equal: Bangkok, Manila, Shenyang, Detroit, and Vancouver. For SOFRID and FORLI 10 (resp. 19) and 6 (resp. 15) airports are associated with R>0.7 (resp. R>0.5). As for the TCC, variabilities (standard deviations) are larger for FORLI than for SOFRID. For Lagos, Shenyang and Bogota, SOFRID provides variabilities less than half of IAGOS and for New-York, Dallas and Vancouver, FORLI's variabilities are more than 1.5 times larger than IAGOS ones. For 7 (resp. 10) airports the ratios of standard deviations are between 0.9 and 1.1 for FORLI (resp. SOFRID). For 4 (resp. 9) airports

these ratios range from 0.7 to 0.9 and from 1.1 to 1.3 for 9 (resp. 8) airports for FORLI (resp. SOFRID). For the remaining airports, standard deviations ratios are less than 0.7 (5 for SOFRID and 3 for FORLI) or higher than 1.3 (1 for SOFRID and 10 for FORLI). Therefore SOFRID reproduces again slightly better the phase of the temporal variations of the surface-600 hPa CO column. The low sensitivities of the retrieval algorithms in the lowermost layers documented by the DFSs (Table 1) and AKs (Fig. 3) explain the lower level of agreement with the IAGOS data for the lower tropospheric column than for the TCC.


For the mid-upper tropospheric column (600-200 hPa) the Taylor indicators are more compact than for the lower tropospheric column (Fig. 5 (a) and (d)) with for instance RMSDs roughly ranging from 0.5 to 1.0 times IAGOS standard deviations for both SOFRID and FORLI. For both algorithms, the Pearson's coefficients associated with this partial column are generally similar or larger than the ones associated with the TCC. The range of altitude between 600 and 200 hPa indeed corresponds to the

maximum sensitivity of the algorithms, as shown by the AKs in Fig. 3. For SOFRID (FORLI), 12 (12) airports are associated with R>0.7 and 14 (9) with 0.6<R<0.7. For 11 airports SOFRID and FORLI's Pearson's coefficients are equal or nearly equal: Lagos, Bangkok, Manila, Kuwait City, Dallas, Nagoya, Philadelphia, Madrid, Detroit, Frankfurt, and Düsseldorf. The ratios





of standard deviations of the retrievals relative to IAGOS ones are lower than for the TCC and the lower tropospheric column as is clearly displayed on Fig. 5. These ratios are ranging from 0.53 (Shenyang) to 1.22 (Hong Kong) for FORLI and from

0.38 (Shenyang) to 0.76 (Hong Kong) for SOFRID. Therefore, for only 8 (resp. 0) airports the ratios of standard deviations are between 0.9 and 1.1 for FORLI (resp. SOFRID). For 15 (resp. 2) airports, the ratios are between 0.7 and 0.9 for FORLI (resp. SOFRID) and for 3 (resp. 0) airports the ratios are within 1.1-1.3. For 7 (resp. 31) airports the ratios are less than 0.7 for FORLI (resp. SOFRID). For the remaining 7 airports, FORLI standard deviations are more than 1.3 larger than IAGOS one. Therefore standard deviations are generally higher for FORLI than for SOFRID even if the IASI retrievals both underestimate

the amplitude of the IAGOS CO variability. As for the TCC, SOFRID slightly better reproduces the phase of the variations with larger Pearson coefficients than FORLI.

The smoothing of the IAGOS profiles by the retrieval AKs has the general effect to improve the agreement with larger Pearson's coefficients and more compact clouds of points with standard deviation ratios closer to the 1:1 circle (Fig. A1). For

SOFRID, the smoothing has little effect for the TCC (Table 2) and lower tropospheric columns (Table A2) but improves significantly the correlations for the mid-upper tropospheric columns (Table A3). For FORLI the variability ratios clearly decrease and come closer to 1 and the Pearson's coefficients clearly increases for the 3 columns.

### 3.3.2   Biases

The biases and corresponding RMSDs for comparisons with raw and smoothed IAGOS data are reported in Table 2 for the TCC and in Table A2 for the surface-600 hPa and Table A3 for the 600-200 hPa partial columns. The median together with the 25th and 75th percentiles of the differences are displayed for FORLI and SOFRID with raw and smoothed IAGOS data and the three columns on Fig. 6 (airports are listed in ascending order of latitude).

For both SOFRID and FORLI the TCC biases at the 33 selected airports are mostly negatives with mean values comprised between -25 and 3% (median differences between -20 and 7%, Fig. 6), and an average over all airports of less than 10% in absolute value for both algorithms. For 24 (14) and 26 (18) airports biases are less than or equal to 10% (5%) in absolute value for FORLI and SOFRID, respectively. Globally, absolute values of FORLI biases are higher than SOFRID ones at a majority (21) of the 33 airports but the global mean biases of both retrievals are not significantly different. The largest negative ($\leq$10%)

TCC biases common to both products occur south of Bangkok (Bogota, Lagos, Ho Chi Minh City) and north of Philadelphia (Madrid, Shenyang, Vancouver). At 9 out of 12 airports south of Taipei, SOFRID negative biases are larger than FORLI's in absolute value. On the contrary, north of Philadelphia, FORLI biases are systematically larger than SOFRID ones. These latitudinal behaviours are clearly visible on Fig. 6. The largest discrepancies (>10% in absolute value) between the two products occur at Bogota, Addis Ababa (SOFRID's absolute value higher), and Toronto (FORLI's absolute value higher). In most cases

the differences between raw and smoothed IAGOS data (Table 2) are not significant for the TCC. Over Lagos and Ho Chi Minh



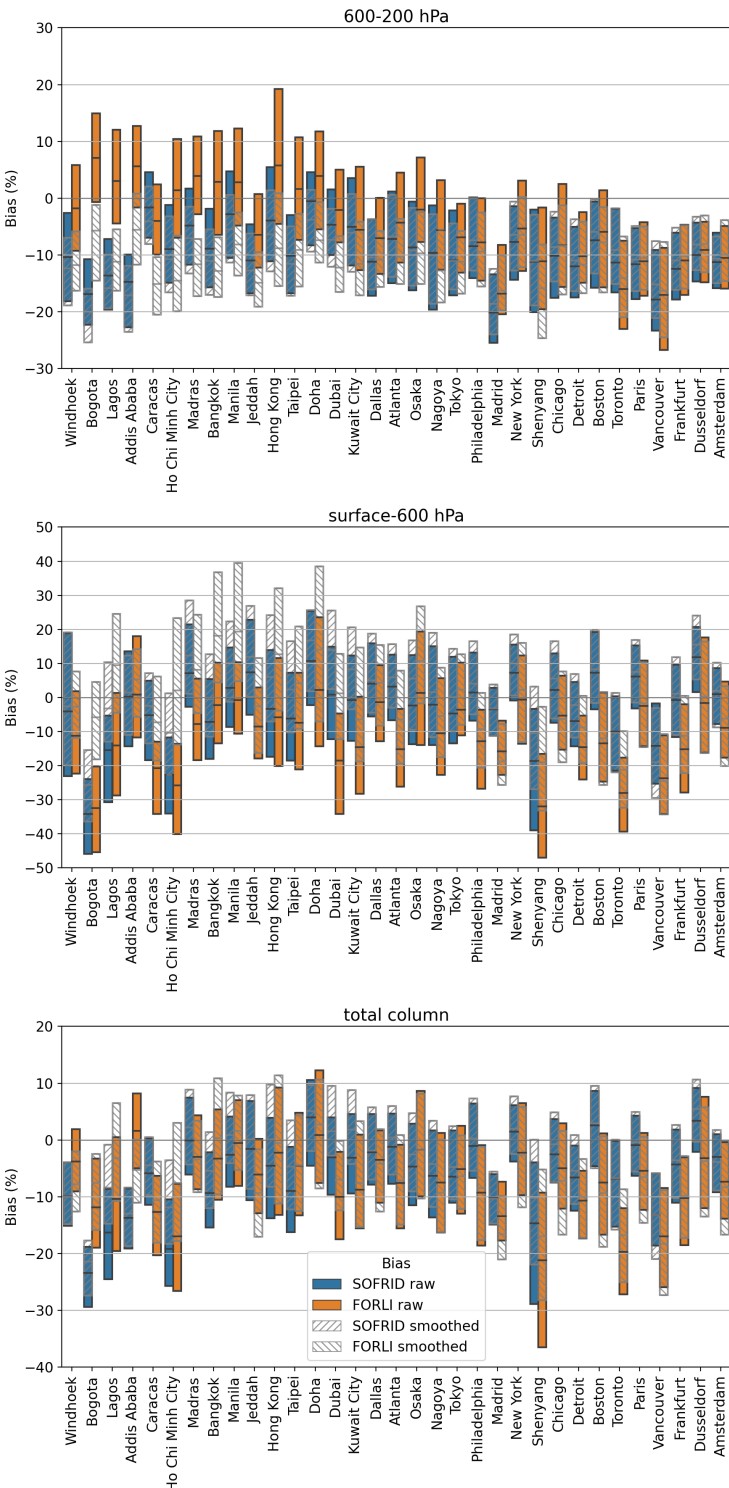

**Figure 6.** Median, 25th and 75th percentiles of the biases between IASI and IAGOS CO columns at the 33 selected airports for the 600-200 hPa (top) surface-600 hPa (middle) and total (bottom). (blue) SOFRID and (orange) FORLI versus IAGOS. (hatched clockwise) SOFRID and (hatched anti-clockwise) FORLI versus IAGOS smoothed. Airports are listed in ascending order of latitude.





City where the negative biases are large the smoothing significantly reduces the biases.

In the surface-600 hPa layer, the biases are mostly negative, ranging from -39 to 10% with mean biases over all the airports of -4% (resp. -11%) for SOFRID (resp. FORLI) (Table A2). For 17 (13) and 27 (15) airports the absolute biases remain below 10% (5%) for FORLI and SOFRID. So SOFRID, and FORLI to a lesser extent, keep low biases for the lower tropospheric column. The latitudinal behaviour of both products is very similar to the one of the TCC with larger negative biases south of Bangkok and north of Philadelphia as can be seen in Fig. 6. For 10 out of the 13 airports north of Philadelphia FORLI absolute biases are larger than SOFRID ones. As could be expected from the information content analysis, smoothing has a larger impact on this lower tropospheric column (see section 3.1). This is especially noticeable for FORLI over Bogota, Caracas, Ho Chi Minh City, Dubai, Kuwait City, and Shenyang where absolute biases are more than 10% smaller when IAGOS data are smoothed. The same is true for SOFRID over Bogota, Lagos, Ho Chi Minh City, and Shenyang.

In the mid-tropospheric layer the median biases are roughly comprised between -20 and 8% (Fig. 6). Instead of the lower tropospheric column, the mean bias over the whole dataset is larger for SOFRID (-11%) than for FORLI (-6%) (Table A3). For 26 (14) and 16 (6) airports the mean absolute biases are less than 10% (5%) for FORLI and SOFRID. SOFRID biases are consistently negative with almost no difference between raw and smoothed IAGOS data. For FORLI, the biases are oscillating around zero, and are mostly positive south of Doha and become significantly negative north of Nagoya. Therefore, as for the TCC, the largest discrepancies between SOFRID and FORLI occur at low latitudes with SOFRID absolute biases larger than FORLI's at 9 out of 12 airports south of Taipei. For FORLI, the application of the AKs brings the biases to large negative values south of Kuwait City and makes little difference for airports further north.

From the comparative analysis of the biases for the three different columns we can conclude that the larger TCC negative biases of SOFRID relative to FORLI south of Taipei are related to the mid-upper troposphere. Conversely, the larger TCC negative biases of FORLI north of Philadelphia are mostly linked to the lower troposphere. The lower impact of the AK smoothing on SOFRID comparisons result from the larger DFS for SOFRID retrievals (Table 1).

## 3.4 Time series

In order to have a better insight into the discrepancies between IASI retrievals and IAGOS data we have plotted the time series of the two datasets and of their differences for coincident dates at 6 airports (Frankfurt, Atlanta, Bangkok, Taipei, Nagoya, Windhoek) selected for their good temporal sampling during the IASI-A period and for their location over different regions.

Frankfurt presents the densest sampling over the whole period with only three periods without observations in 2010, 2014 and 2020 (Fig. 7). As already documented in De Wachter et al. (2012), for SOFRID and FORLI, the TCC biases are negative with a seasonal cycle characterised by large biases in winter-spring and no significant biases in summer (Fig. 7 bottom panel).



The biases are similar for both algorithms during 2008-2010 and 2015-2019 but FORLI displays larger negative biases for the period 2011-2015. These different behaviours in FORLI retrievals can be related to the two major updates of EUMETSAT Level 2 data processing that occured in September-December 2010 and end of September 2014 according to Van Damme et al. (2017) (see Table 2).

The same behaviours are observed for the surface-600 hPa layer with larger biases variations from -40% in winter to 20% in summer (Fig. 7 middle panel). The larger biases in winter are related to the lower sensitivity to the lower troposphere when the surface is cold and the surface atmosphere thermal contrast low as detailed in section 3.1. As for the TCC the SOFRID and FORLI biases are similar except for the period 2011-2015. During this period FORLI's biases are about 20% lower than SOFRID and remain negatives during all seasons when SOFRID biases become positive in summer. On average, FORLI un-

derestimates IAGOS lower tropospheric columns by 16% against 3% for SOFRID (Table A2). In the 600-200 hPa layer, the biases of both algorithms display less seasonal variability with values in the (-20; 0%) range (Fig. 7 top panel) and very similar mean biases of -11 and -13% for FORLI and SOFRID respectively (Table A3).

        Atlanta provides less data than Frankfurt but displays the same behaviour (Fig. 8). FORLI is underestimating the TCC up

to 20% with an average of -8% and SOFRID biases are oscillating around zero with an average of -2%. The same is true in the lower troposphere with mean biases of -14 and 2% for FORLI and SOFRID. In the mid troposphere, both retrievals are in better agreement with similar biases of -5 and -8% for FORLI and SOFRID. The seasonal and interannual bias variations are not really clear compared to Frankfurt due to the lower temporal sampling.

Over Bangkok (Fig. 9), valid data are provided mostly from 2015 to 2018. The IASI retrieved TCC correctly capture the seasonal variations from IAGOS with winter spring maxima and summer minima (Fig. 9 bottom). SOFRID underestimates IAGOS by up to 20% with an average bias of 10% and FORLI oscillates between -20 and 20% with a mean bias of -2%. When the AKs are applied to the IAGOS profiles, little differences are observed from comparisons with raw IAGOS data for SOFRID. For FORLI, the overestimation is slightly higher when the IAGOS profiles are smoothed.


        For the surface-600 hPa layer, the general behaviour is similar to the TCC with larger bias variations. For raw IAGOS data both SOFRID and FORLI biases are roughly within the -20; 20% boundaries but SOFRID's mean bias (-8%) is larger than FORLI's (-1%). Application of the AKs smoothing leads to large overestimation of FORLI retrievals from 20 to 60%. In the mid-upper tropospheric layer (600-200 hPa), the seasonal variability is lower and SOFRID (resp. FORLI) underestimates

(resp. overestimates) IAGOS by up to 20% with a mean bias of -10% (resp. 3%). AKs smoothing changes little for SOFRID retrievals. For FORLI the application of the AKs leads to an underestimation compensating the overestimation in the lower layer.

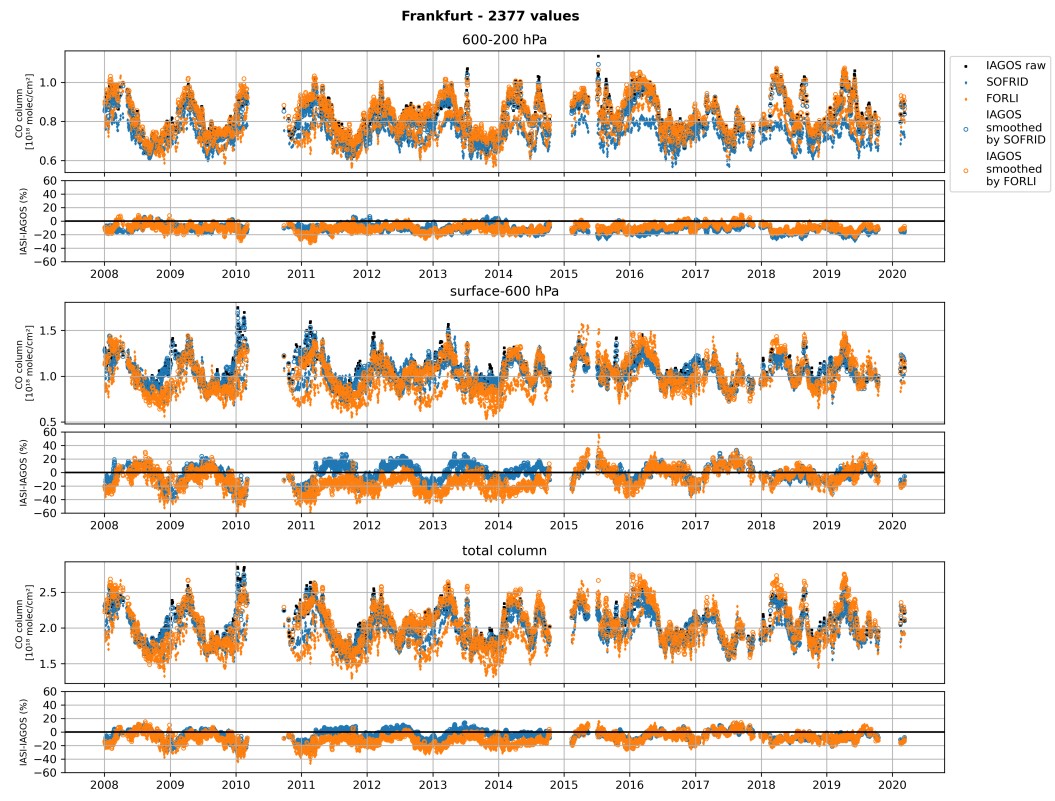

**Figure 7.** Time series of SOFRID (blue diamonds), FORLI (orange diamonds), IAGOS raw (black diamonds), IAGOS smoothed with SOFRID AKs (empty blue circles) and IAGOS smoothed with FORLI AKs (empty orange circles) CO columns at Frankfurt. (Upper panels) 600-200 hPa, (middle panels) surface-600 hPa and (lower panels) total columns. The lower panels display the differences between SOFRID and raw IAGOS (blue diamonds), FORLI and raw IAGOS (orange diamonds), SOFRID and smoothed IAGOS (empty blue circles), FORLI and smoothed IAGOS (empty orange circles).

Over Taipei, data are available for a short period in 2012-2013 and from 2015 to 2018 with a denser sampling than over
Bangkok. The seasonal TCC variations are characterised by marked spring maxima and both SOFRID and FORLI display a very good and similar agreement with mean biases within -20; 0% except in 2018 where they become positive but remain below 20%. The mean bias relative to IAGOS columns is larger for SOFRID (-9%) than for FORLI (-4%). Contrarily to Frankfurt, the biases do not display clear seasonal cycles. In the surface-600 hPa layer, the variations are also captured by both algorithms with more variable biases than for the TCC. The biases are mostly in the -20; 20% range until the spring of 2018 where they
become positive and remain below 40%. In the 600-200 hPa layer SOFRID underestimates IAGOS by up to 20% for the spring maxima and FORLI generaly displays a better agreement especially in representing the maxima.

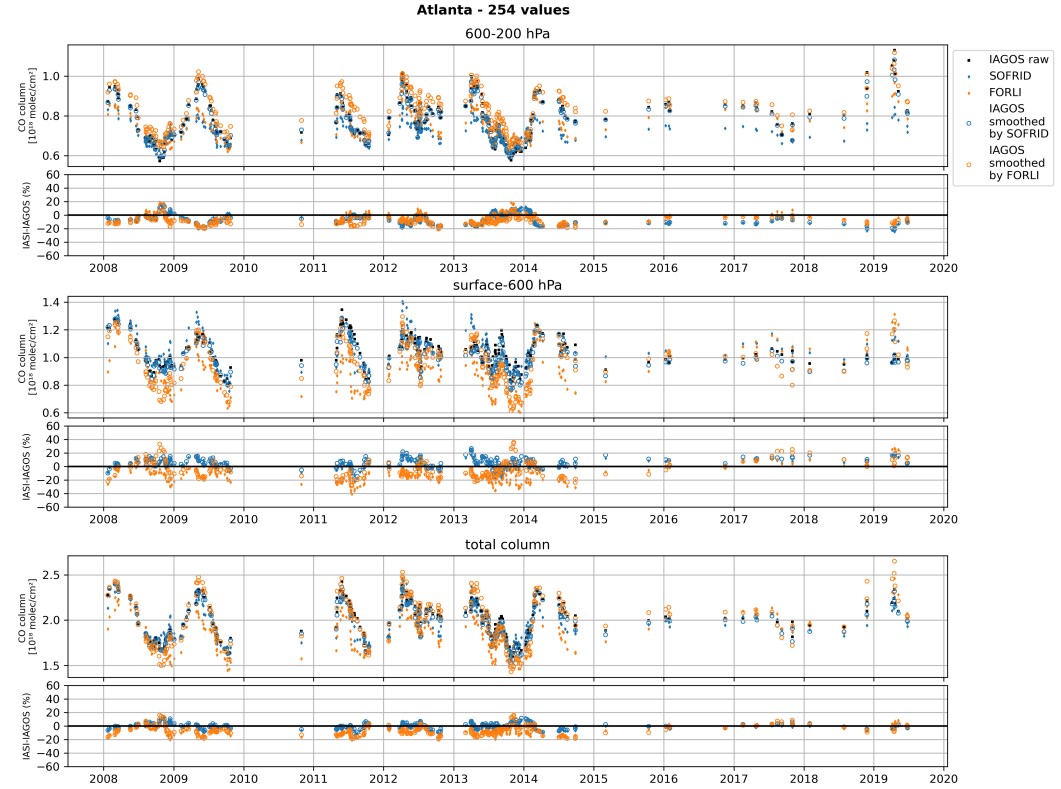

**Figure 8.** Same as Fig. 7 for Atlanta.

Nagoya IAGOS data date back to the end of 2010 but are much sparser than over Taipei. The years with the best sampling are 2011-2013. Afterwards data are too sparse to document the seasonal variability. For the TCC, FORLI and SOFRID display

the largest biases (up to -30% for FORLI) in winter-spring and better agreement in summer. For the surface-600 hPa column, the biases are negative in winter-spring and positive in summer and FORLI's underestimation (-11%) is larger than SOFRID's (-3%). In the 600-200 hPa layer, the bias seasonal variations are less important and FORLI's bias (-7%) is lower than SOFRID's (-13%).

Windhoek is an interesting location to document the ability of IASI retrievals to capture the impact of biomass burning fire plumes on the CO profiles (Fig. 12). In De Wachter et al. (2012), FORLI and SOFRID retrievals were compared to Windhoek IAGOS data for 2008-2009. Here we have data from 2011 to 2013 to improve the comparisons. Both algorithms capture the large spring biomass burning maxima visible over the three layers. The underestimation of TCC by the retrievals is larger for SOFRID (-11%) than for FORLI (-4%). As for the other locations, in the surface-600 hPa layer, the biases are larger and display

a stronger seasonal cycle. The SOFRID biases are negative in boreal summer and positive in winter at the end of the biomass burning season and smaller on average (-7%) than FORLI's (-11%). The positive biases in the lower layer are compensated by





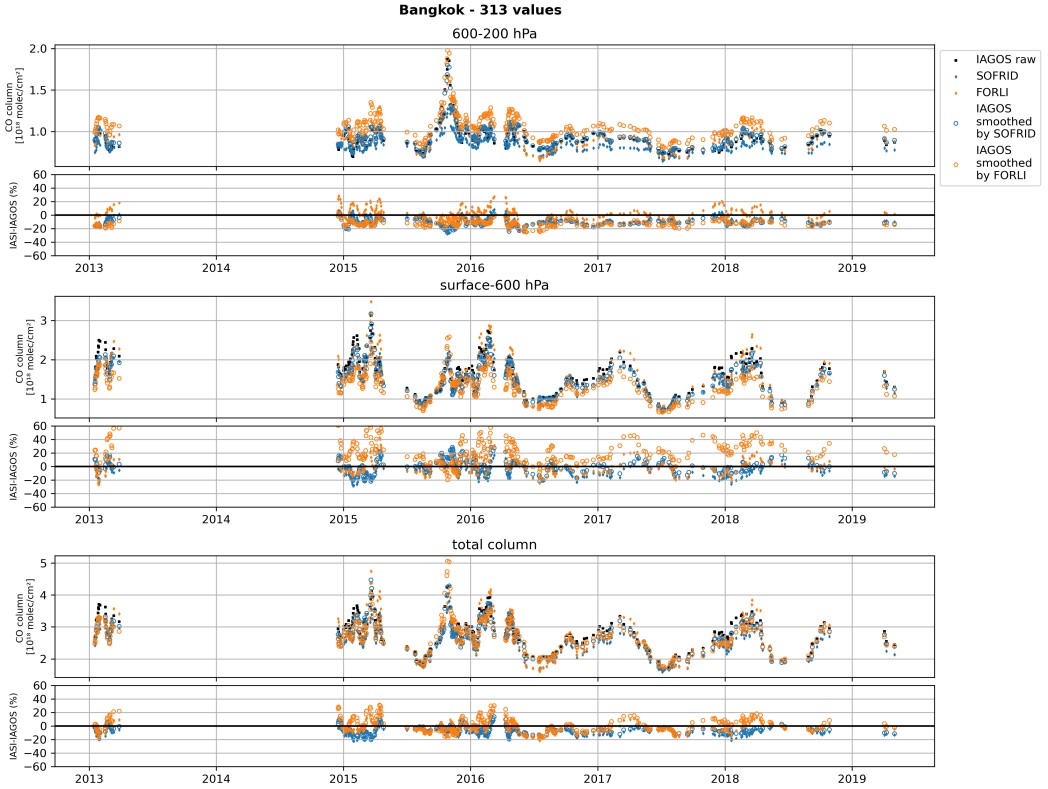

**Figure 9.** Same as Fig. 7 for Bangkok.

an important underestimation in the upper layer. This effect is less noticeable for FORLI.

## 4 Conclusions

We have used data from the IAGOS European Research Infrastructure to validate CO IASI columns retrieved from the SOFRID and FORLI algorithms over the whole Metop-A period (2008-2020). Only airports providing at least 60 days with valid data have been selected resulting in 14211 profiles (8478 days) for 33 airports. From an analysis of the information content of both retrieval algorithms, we have chosen to make comparisons for the total column of CO (TCC), the lower tropospheric (surface-600 hPa) and the mid-upper tropospheric (600-200 hPa) partial columns.

SOFRID and FORLI have slightly different behaviours concerning the reproduction of the CO variations. For the TCC and the surface-600 hPa column SOFRID provides larger correlation coefficients for a majority (29) of the 33 airports, meaning a better agreement for the phase of IAGOS CO temporal variations for these columns. For the 600-200 hPa partial column the correlation coefficients are closer for both algorithms with larger coefficients computed for SOFRID at only 18 airports.

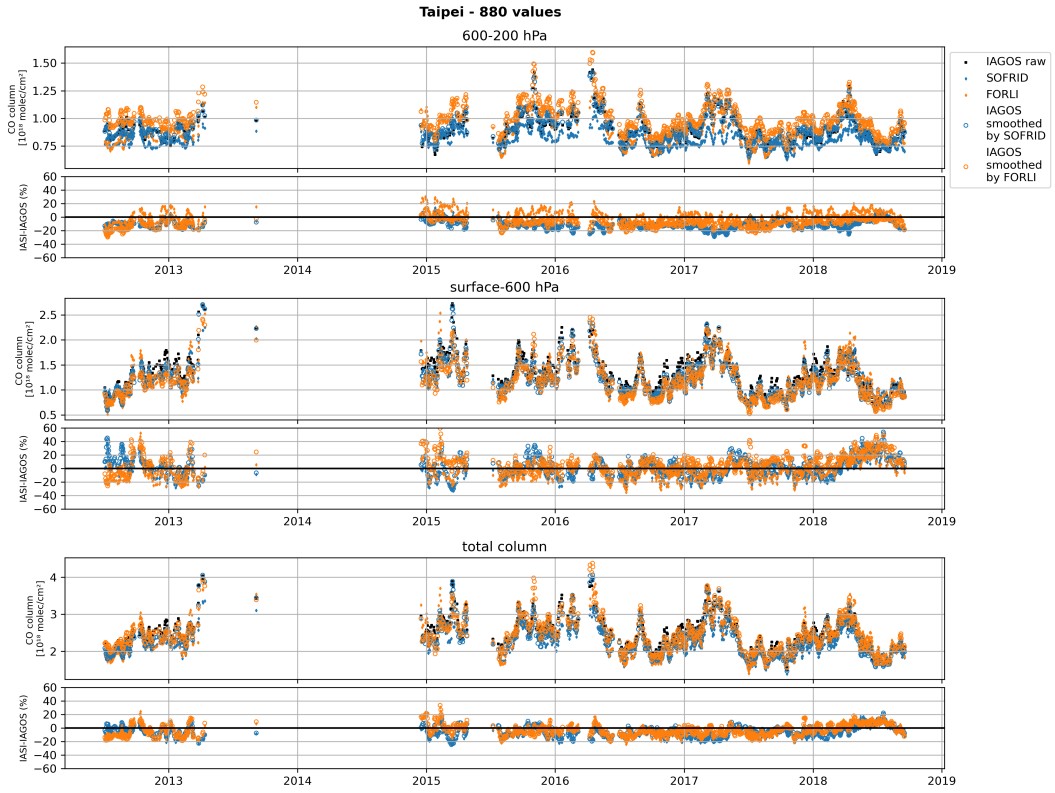

**Figure 10.** Same as Fig. 7 for Taipei.

Concerning the variability of the TCC, the standard deviations are close to IAGOS ones (ratios within 0.7-1.3) at a majority (29
for SOFRID and 28 for FORLI) of the airports. FORLI (resp. SOFRID) is generally overestimating (underestimating) IAGOS
variabilities for the three layers. For the lower troposphere standard deviations ratios are within 0.7-1.3 for 27 (resp. 20) of the
airports for SOFRID (resp. FORLI). For the mid-upper troposphere FORLI variabilities are in good agreement (ratios within
0.7-1.3) with IAGOS for most (26) of the airports. SOFRID is underestimating the mid-upper tropospheric CO variability (with
ratios lower than 0.7) at a majority (31) of airports.

On average over all the dataset, SOFRID (resp. FORLI) underestimates IAGOS TCC by 6±14% (resp 8±16%) with a corre-
lation coefficient of 0.81 (resp. 0.78). For both algorithms, the biases are not geographically uniforms. At 9 out of 12 airports,
south of Taipei SOFRID's TCC negative biases are larger in absolute value and, north of Philadelphia (13 airports) FORLI's
underestimations are larger. The larger SOFRID TCC biases result mainly from large biases in the mid-upper troposphere.
SOFRID average bias in the 600-200 hPa layer (-11±13%) is about twice larger than FORLI's (-6±15%). The larger FORLI
TCC biases are mainly related to the large biases of FORLI in the lower troposphere. Indeed, FORLI's mean bias (-11±27%)
is almost 3 times larger than SOFRID's (-4±24%) in the surface-600 hPa layer.

Data from Frankfurt which is the airport with the densest and longest IAGOS timeserie show that IASI retrievals allow to cap-



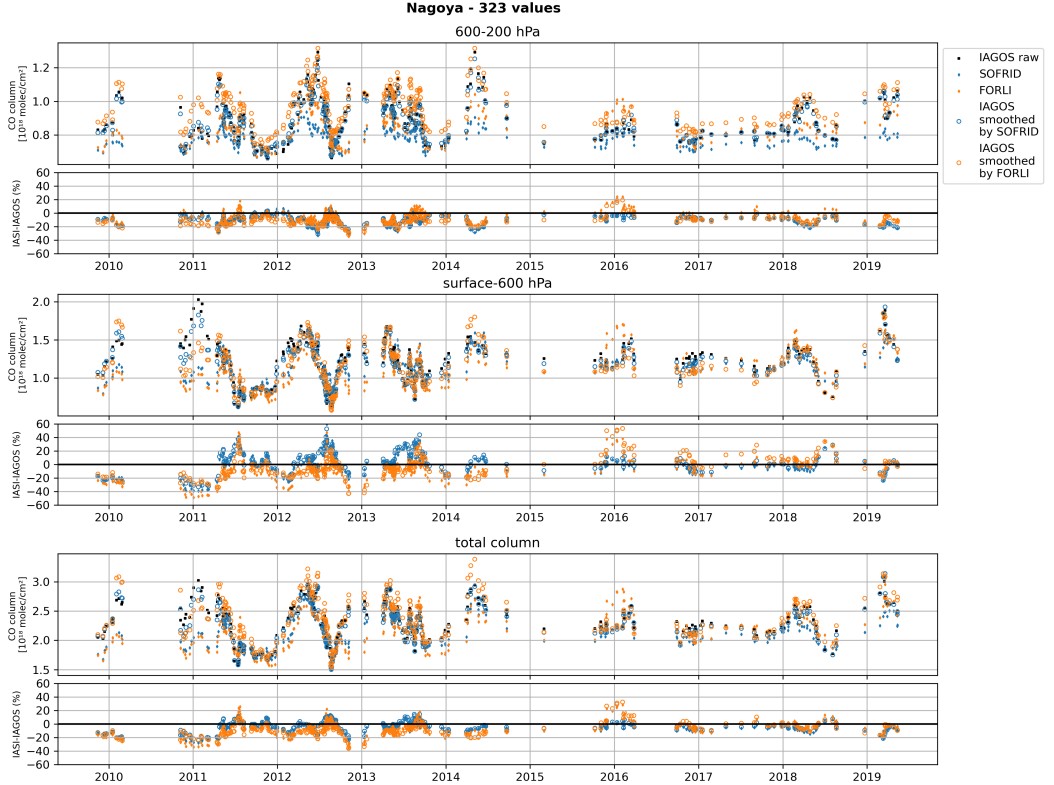

**Figure 11.** Same as Fig. 7 for Nagoya.

ture correctly the seasonal variations of TCC with the summer maxima and winter-spring minima. Nevertheless, both retrievals display an important underestimation in winter-spring and almost no bias in summer and FORLI's biases are significantly

larger during the 2011-2015 period. This can be explained by version changes in EUMETSAT Level 2 data processing. Inspection of the partial columns time series highlights that the temporal variability of the TCC biases are mostly stemming from the surface-600 hPa columns. For Taipei which is the second airport with the longest and densest IAGOS dataset, there is no clear seasonal variations of the biases for the 3 different columns. At Windhoek, IASI retrievals are able to capture the large TCC maxima in austral spring when biomass burning are active over southern Africa. SOFRID tends to underestimate CO and

especially the impact of biomass burning in the mid-upper troposphere.

To conclude, SOFRID and FORLI are able to capture the TCC spatio-temporal variability over the 12 years of Metop-A with an underestimation of less than 8% not statistically significant. Nevertheless, this average figure does not represent a homogeneous reality and we have shown that the IAGOS database highlighted the relative strengths and weaknesses of both retrievals to capture the 4D variations of CO.




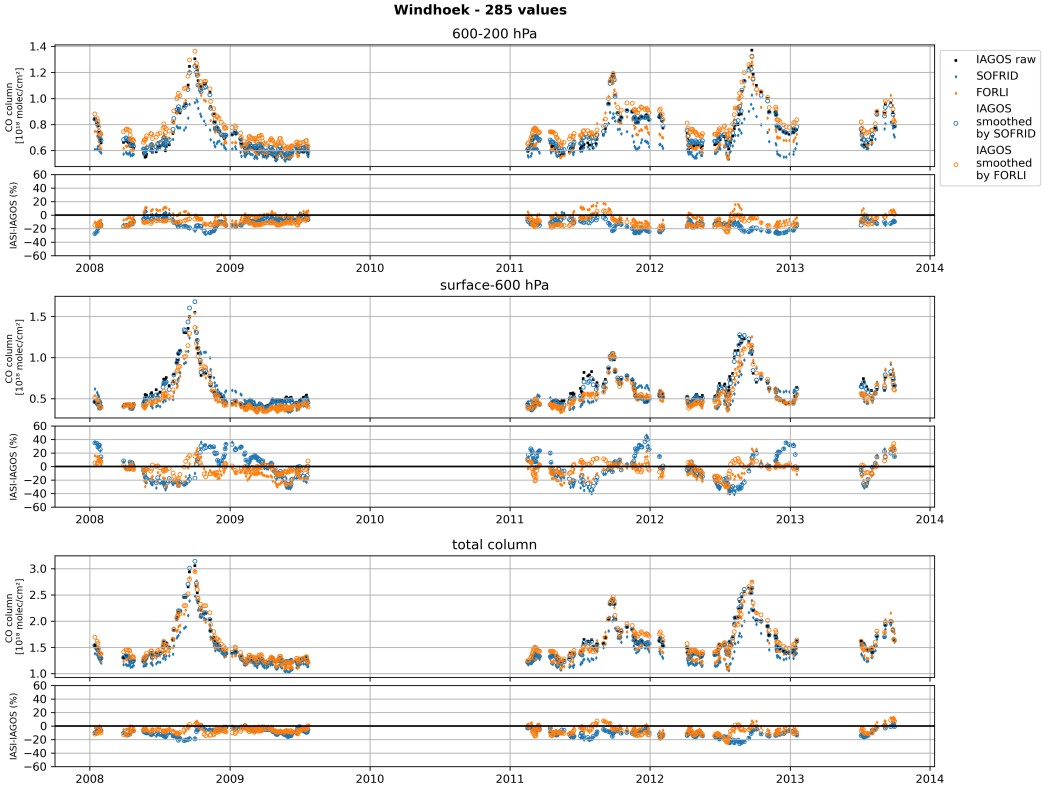

**Figure 12.** Same as Fig. 7 for Windhoek.

*Data availability.* IAGOS data are available at https://www.iagos.fr

SOFRID CO daily and monthly data are available for the whole period through the Service de données de l'Observatoire Midi-Pyrénées (https://iasi-sofrid.sedoo.fr/)

FORLI CO products are available at https://navigator.eumetsat.int/product/EO:EUM:DAT:METOP:IASIL2COX from May 14, 2019, and
through the AERIS data infrastructure (http://iasi.aeris-data.fr/co/) for the whole IASI observation period.

*Author contributions.* BB is responsible for SOFRID and is the PI of this study.

ELF is in charge of SOFRID operations and development.

PL and YB were in charge of the realisation of the validation scripts and they participated to the redaction of the paper.

JHL is in charge of the FORLI processing and she participated to the redaction of the paper.

DH is responsible of the development of the FORLI algorithm.

BS has provided IAGOS data and expertise and participated to the redaction of the paper.



| Airport | Latitude | Longitude | Number of IAGOS days |
|---|---|---|---|
| Windhoek | -22.49 | 17.46 | 285 |
| Bogota | 4.71 | -74.16 | 103 |
| Lagos | 6.58 | 3.31 | 74 |
| Addis Ababa | 8.98 | 38.80 | 90 |
| Caracas | 10.60 | -67.00 | 116 |
| Ho Chi Minh City | 10.82 | 106.67 | 97 |
| Madras | 13.01 | 80.22 | 168 |
| Bangkok | 13.57 | 100.71 | 313 |
| Manila | 14.53 | 121.06 | 66 |
| Jeddah | 21.67 | 39.14 | 84 |
| Hong Kong | 22.31 | 113.93 | 373 |
| Taipei | 25.09 | 121.24 | 880 |
| Doha | 25.25 | 51.57 | 67 |
| Dubai | 25.32 | 55.53 | 93 |
| Kuwait City | 29.23 | 47.97 | 82 |
| Dallas | 32.90 | -97.05 | 118 |
| Atlanta | 33.63 | -84.43 | 254 |
| Osaka | 34.51 | 135.25 | 110 |
| Nagoya | 34.85 | 136.81 | 323 |
| Tokyo | 35.76 | 140.38 | 111 |
| Philadelphia | 39.86 | -75.29 | 191 |
| Madrid | 40.49 | -3.55 | 119 |
| New York | 40.69 | -74.17 | 202 |
| Shenyang | 41.64 | 123.48 | 100 |
| Chicago | 41.98 | -87.93 | 146 |
| Detroit | 42.23 | -83.35 | 99 |
| Boston | 42.37 | -71.00 | 142 |
| Toronto | 43.68 | -79.63 | 164 |
| Paris | 49.00 | 2.56 | 637 |
| Vancouver | 49.19 | -123.19 | 180 |
| Frankfurt | 50.04 | 8.56 | 2377 |
| Dusseldorf | 51.28 | 6.76 | 227 |
| Amsterdam | 52.30 | 4.74 | 87 |

**Table A1.** Latitude, longitude and number of days with valid IAGOS profiles at the 33 selected airports.



| Airport | FORLI | | | | SOFRID | | | |
|---|---|---|---|---|---|---|---|---|
| | Raw | IAGOS | Smoothed | IAGOS | Raw | IAGOS | Smoothed | IAGOS |
| | R | Bias % | R | Bias % | R | Bias % | R | Bias % |
| Windhoek | 0.84 | -11 ± 25 | 0.92 | -2 ± 18 | 0.71 | -7 ± 34 | 0.76 | -5 ± 33 |
| Bogota | 0.36 | -38 ± 34 | 0.70 | -7 ± 16 | 0.41 | -39 ± 33 | 0.58 | -28 ± 25 |
| Lagos | 0.75 | -14 ± 31 | 0.81 | 14 ± 35 | 0.82 | -23 ± 30 | 0.90 | -5 ± 20 |
| Addis Ababa | 0.29 | -3 ± 37 | 0.70 | 4 ± 18 | 0.52 | -6 ± 31 | 0.64 | -2 ± 24 |
| Caracas | 0.43 | -23 ± 22 | 0.50 | -7 ± 23 | 0.61 | -8 ± 19 | 0.62 | -5 ± 18 |
| Ho Chi Minh City | 0.65 | -28 ± 26 | 0.70 | 3 ± 27 | 0.72 | -26 ± 26 | 0.76 | -12 ± 22 |
| Madras | 0.67 | -5 ± 21 | 0.73 | 10 ± 22 | 0.71 | 9 ± 19 | 0.75 | 13 ± 19 |
| Bangkok | 0.82 | -1 ± 22 | 0.76 | 21 ± 31 | 0.83 | -8 ± 19 | 0.83 | -1 ± 19 |
| Manila | 0.70 | -1 ± 18 | 0.74 | 21 ± 21 | 0.71 | 3 ± 19 | 0.76 | 11 ± 19 |
| Jeddah | 0.39 | -7 ± 19 | 0.56 | -3 ± 21 | 0.31 | 9 ± 21 | 0.38 | 13 ± 19 |
| Hong Kong | 0.69 | -4 ± 29 | 0.68 | 14 ± 35 | 0.71 | -4 ± 23 | 0.72 | 6 ± 24 |
| Taipei | 0.74 | -7 ± 25 | 0.79 | 6 ± 26 | 0.80 | -7 ± 22 | 0.81 | 1 ± 22 |
| Doha | 0.08 | 1 ± 25 | 0.11 | 13 ± 30 | 0.14 | 10 ± 23 | 0.27 | 15 ± 20 |
| Dubai | 0.15 | -19 ± 23 | 0.39 | 2 ± 21 | 0.22 | -0 ± 21 | 0.26 | 13 ± 18 |
| Kuwait City | 0.32 | -13 ± 21 | 0.43 | -2 ± 23 | 0.29 | -0 ± 20 | 0.30 | 9 ± 20 |
| Dallas | 0.59 | -1 ± 20 | 0.66 | 1 ± 21 | 0.64 | 5 ± 16 | 0.63 | 6 ± 16 |
| Atlanta | 0.50 | -14 ± 20 | 0.59 | -6 ± 20 | 0.62 | 2 ± 15 | 0.65 | 5 ± 15 |
| Osaka | 0.37 | 3 ± 27 | 0.52 | 12 ± 26 | 0.42 | -3 ± 22 | 0.46 | 2 ± 22 |
| Nagoya | 0.55 | -11 ± 28 | 0.62 | -6 ± 27 | 0.62 | -3 ± 26 | 0.64 | 0 ± 25 |
| Tokyo | 0.47 | -3 ± 20 | 0.61 | 2 ± 19 | 0.43 | -4 ± 18 | 0.49 | -0 ± 17 |
| Philadelphia | 0.56 | -15 ± 18 | 0.65 | -11 ± 19 | 0.62 | 3 ± 16 | 0.62 | 5 ± 17 |
| Madrid | 0.50 | -13 ± 15 | 0.52 | -16 ± 16 | 0.72 | -4 ± 12 | 0.65 | -4 ± 13 |
| New York | 0.43 | -1 ± 22 | 0.53 | 0 ± 23 | 0.65 | 7 ± 15 | 0.64 | 9 ± 16 |
| Shenyang | 0.41 | -34 ± 38 | 0.45 | -18 ± 35 | 0.42 | -27 ± 37 | 0.41 | -17 ± 35 |
| Chicago | 0.47 | -4 ± 21 | 0.46 | -6 ± 23 | 0.44 | 2 ± 17 | 0.43 | 3 ± 19 |
| Detroit | 0.50 | -14 ± 16 | 0.63 | -11 ± 17 | 0.50 | -6 ± 14 | 0.50 | -3 ± 15 |
| Boston | 0.36 | -10 ± 26 | 0.34 | -12 ± 28 | 0.44 | 8 ± 17 | 0.44 | 9 ± 18 |
| Toronto | 0.25 | -28 ± 18 | 0.49 | -22 ± 20 | 0.32 | -11 ± 17 | 0.25 | -11 ± 18 |
| Paris | 0.43 | -2 ± 20 | 0.57 | -3 ± 19 | 0.56 | 5 ± 15 | 0.56 | 7 ± 14 |
| Vancouver | 0.71 | -18 ± 24 | 0.75 | -21 ± 22 | 0.72 | -11 ± 23 | 0.66 | -15 ± 24 |
| Frankfurt | 0.32 | -16 ± 22 | 0.56 | -11 ± 19 | 0.45 | -3 ± 18 | 0.41 | -2 ± 18 |
| Dusseldorf | 0.11 | -0 ± 28 | 0.23 | 1 ± 26 | 0.42 | 9 ± 18 | 0.44 | 13 ± 17 |
| Amsterdam | 0.22 | -7 ± 22 | 0.38 | -7 ± 21 | 0.36 | -1 ± 15 | 0.30 | -0 ± 16 |
| All | 0.74 | -11 ± 27 | 0.75 | -3 ± 27 | 0.76 | -4 ± 24 | 0.77 | 1 ± 22 |

**Table A2.** Pearson coefficients and biases for FORLI and SOFRID for surface-600 hPa columns comparisons with raw and smoothed IAGOS data at the 33 selected airports listed in ascending order of latitude.




| Airport | FORLI | | | | SOFRID | | | |
|---|---|---|---|---|---|---|---|---|
| | Raw | IAGOS | Smoothed | IAGOS | Raw | IAGOS | Smoothed | IAGOS |
| | R | Bias % | R | Bias % | R | Bias % | R | Bias % |
| Windhoek | 0.90 | -2 ± 12 | 0.92 | -11 ± 9 | 0.86 | -13 ± 17 | 0.88 | -15 ± 13 |
| Bogota | 0.71 | 7 ± 11 | 0.72 | -7 ± 10 | 0.73 | -17 ± 10 | 0.71 | -21 ± 9 |
| Lagos | 0.76 | 3 ± 12 | 0.83 | -11 ± 9 | 0.77 | -14 ± 12 | 0.81 | -16 ± 11 |
| Addis Ababa | 0.83 | 5 ± 11 | 0.82 | -6 ± 10 | 0.80 | -17 ± 11 | 0.81 | -18 ± 10 |
| Caracas | 0.66 | -3 ± 11 | 0.72 | -15 ± 9 | 0.70 | -2 ± 10 | 0.72 | -4 ± 9 |
| Ho Chi Minh City | 0.67 | 2 ± 14 | 0.76 | -13 ± 10 | 0.71 | -9 ± 11 | 0.75 | -11 ± 10 |
| Madras | 0.78 | 3 ± 10 | 0.80 | -12 ± 9 | 0.73 | -6 ± 11 | 0.74 | -8 ± 10 |
| Bangkok | 0.75 | 3 ± 15 | 0.82 | -12 ± 11 | 0.75 | -10 ± 15 | 0.78 | -12 ± 13 |
| Manila | 0.81 | 4 ± 13 | 0.89 | -8 ± 9 | 0.82 | -4 ± 12 | 0.84 | -6 ± 11 |
| Jeddah | 0.72 | -7 ± 11 | 0.76 | -15 ± 9 | 0.69 | -11 ± 11 | 0.72 | -13 ± 10 |
| Hong Kong | 0.60 | 8 ± 19 | 0.72 | -7 ± 14 | 0.64 | -4 ± 15 | 0.70 | -6 ± 13 |
| Taipei | 0.68 | 2 ± 15 | 0.76 | -9 ± 11 | 0.66 | -11 ± 14 | 0.69 | -12 ± 13 |
| Doha | 0.54 | 3 ± 11 | 0.61 | -7 ± 9 | 0.61 | -2 ± 10 | 0.63 | -4 ± 9 |
| Dubai | 0.49 | -1 ± 10 | 0.49 | -12 ± 9 | 0.42 | -5 ± 10 | 0.48 | -8 ± 8 |
| Kuwait City | 0.42 | -4 ± 12 | 0.50 | -11 ± 10 | 0.42 | -5 ± 11 | 0.45 | -6 ± 10 |
| Dallas | 0.82 | -8 ± 11 | 0.81 | -11 ± 9 | 0.82 | -12 ± 12 | 0.83 | -11 ± 11 |
| Atlanta | 0.75 | -5 ± 12 | 0.75 | -10 ± 10 | 0.73 | -8 ± 13 | 0.73 | -8 ± 12 |
| Osaka | 0.58 | -0 ± 15 | 0.61 | -8 ± 12 | 0.60 | -9 ± 14 | 0.61 | -9 ± 12 |
| Nagoya | 0.55 | -7 ± 19 | 0.59 | -12 ± 15 | 0.56 | -13 ± 19 | 0.57 | -13 ± 16 |
| Tokyo | 0.53 | -7 ± 15 | 0.56 | -11 ± 13 | 0.60 | -12 ± 14 | 0.60 | -11 ± 12 |
| Philadelphia | 0.69 | -8 ± 12 | 0.70 | -10 ± 10 | 0.69 | -8 ± 12 | 0.70 | -8 ± 11 |
| Madrid | 0.68 | -16 ± 10 | 0.67 | -16 ± 8 | 0.69 | -20 ± 10 | 0.69 | -19 ± 9 |
| New York | 0.75 | -6 ± 14 | 0.76 | -6 ± 11 | 0.84 | -9 ± 14 | 0.83 | -8 ± 13 |
| Shenyang | 0.36 | -14 ± 27 | 0.37 | -18 ± 23 | 0.40 | -15 ± 27 | 0.39 | -15 ± 25 |
| Chicago | 0.55 | -8 ± 14 | 0.54 | -9 ± 12 | 0.62 | -12 ± 13 | 0.62 | -10 ± 12 |
| Detroit | 0.68 | -10 ± 11 | 0.68 | -11 ± 10 | 0.68 | -13 ± 11 | 0.69 | -11 ± 10 |
| Boston | 0.57 | -8 ± 14 | 0.55 | -9 ± 12 | 0.59 | -9 ± 14 | 0.58 | -7 ± 13 |
| Toronto | 0.45 | -17 ± 13 | 0.46 | -15 ± 13 | 0.49 | -9 ± 13 | 0.50 | -8 ± 12 |
| Paris | 0.70 | -11 ± 10 | 0.71 | -11 ± 9 | 0.68 | -12 ± 11 | 0.70 | -11 ± 10 |
| Vancouver | 0.46 | -17 ± 17 | 0.45 | -15 ± 16 | 0.41 | -18 ± 15 | 0.44 | -15 ± 14 |
| Frankfurt | 0.68 | -11 ± 11 | 0.69 | -11 ± 10 | 0.68 | -13 ± 11 | 0.70 | -11 ± 10 |
| Dusseldorf | 0.68 | -10 ± 9 | 0.67 | -8 ± 8 | 0.69 | -10 ± 8 | 0.70 | -8 ± 8 |
| Amsterdam | 0.56 | -11 ± 10 | 0.61 | -10 ± 9 | 0.60 | -12 ± 9 | 0.64 | -11 ± 8 |
| All | 0.68 | -6 ± 15 | 0.81 | -11 ± 11 | 0.71 | -11 ± 13 | 0.75 | -11 ± 12 |

**Table A3.** Pearson coefficients and biases for FORLI and SOFRID for 600-200 hPa columns comparisons with raw and smoothed IAGOS data at the 33 selected airports listed in ascending order of latitude.



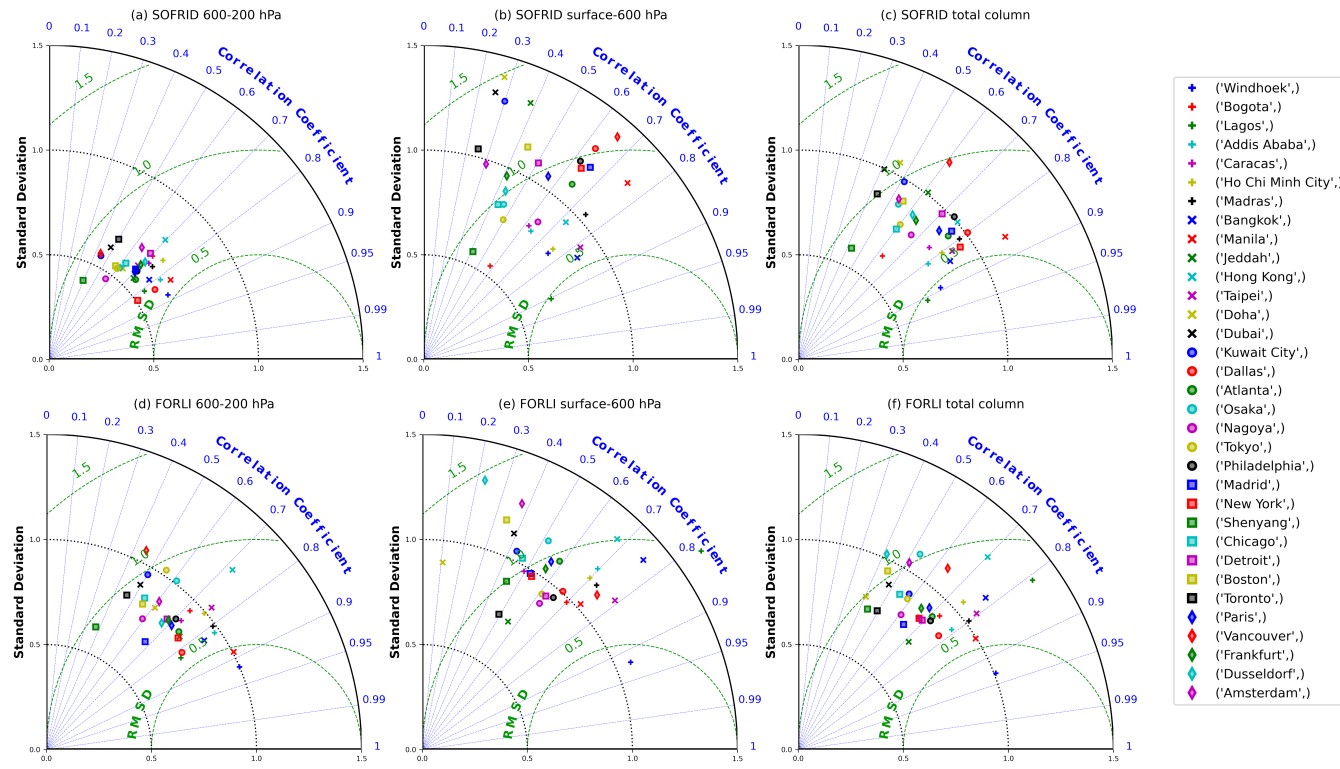

**Figure A1.** Taylor diagrams for the SOFRID (top) and FORLI (bottom) versus IAGOS smoothed data comparisons for the 600-200 hPa (left)**, surface-600** hPa (middle) and total (right) CO columns.

*Competing interests.* We have no competing interests.

*Acknowledgements.* The IASI mission is a joint mission of Eumetsat and the Centre National d'Etudes Spatiales (CNES, France). The IASI L1 and L2 data are distributed in near real time by Eumetsat through the Eumetcast system distribution. The authors acknowledge the AERIS data infrastructure (https://www.aeris-data.fr) for providing access to the IASI Level 1 radiance and to the IASI Level 2 data. This research was carried out in the framework of the IASI-Chimie project supported by the TOSCA/CNES program which financed Pierre Loicq. MOZAIC/CARIBIC/IAGOS data were created with support from the European Commission, national agencies in Germany (BMBF), France (MESR), and the UK (NERC), and the IAGOS member institutions (http://www.iagos.fr/partners). The participating airlines (Lufthansa, Air France, Austrian, China Airlines, Hawaiian Airlines, Air Canada, Iberia, Eurowings Discover, Cathay Pacific, Air Namibia, Sabena) supported IAGOS by carrying the measurement equipment free of charge since 1994. The data are available at http://www.iagos.fr thanks to additional support from AERIS.





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
