# Peer review of "Validation of 12 years (2008-2019) of IASI-A CO with IAGOS aircraft observations"

_EGUsphere, 2024_

## Author Comment (AC1)

**Replies to RC2**: 'Comment on egusphere-2024-30', Anonymous Referee #2, 07 May 2024

We thank both reviewers for their thorough reading of the paper and their appropriate suggestions, which helped us improve our manuscript.

Black: reviewer comment
*Blue: authors reply*
*Red: manuscript correction*

**General comments:**

While the authors achieve to provide an extensive validation and present datasets which reveal relevant information, the manuscript also show some weaknesses:

First, some information is missing about the details of the IASI retrievals and its development over time. There are several quality and error flags as well as data processing updates mentioned without explanation and discussion of potential impact on the validation.

*SOFRID and FORLI CO retrievals did not undergo important modifications since they were first published with De Wachter et al. (2012) and Hurtmans et al. (2012).*

*We provide and document here the first important update of SOFRID-CO (RT code, spectral window, retrieval noise) since De Wachter et al. (2012). To document the impact of SOFRID updates on the results would make the paper difficult to read: we already have 2 products in 3 layers with raw and smoothed data at 33 airports. The former product has been validated in De Wachter et al. (2012) and the reader can find the results there.*

*Concerning FORLI, the retrieval is very similar to the one documented in Hurtmans et al. (2012) and validated in De Wachter et al. (2012). See reply to RC1 l80: "FORLI v20151001 is mainly a technical and spectroscopy update. We have added some details to the text of the article…"*

It is also not clear what IASI-B/C will provide and if there are any differences to IASI-A.

*The paper only deals with IASI-A. We have modified the title accordingly: "Validation of 12 years (2008-2019) of IASI-A CO with IAGOS aircraft observations". Validation of IASI-B and C is of course of great interest but the focus here is on the validation of 2 retrievals with different algorithms.*

*As some comparisons have been made between IASI-A, B and C in a EUMETSAT report concerning FORLI validation with NDACC data, we have added the main conclusions of the report concerning the 3 sensors and the reader is referred to the report for details (see reply to reviewer #1 general comment).*

It would also be intriguing to learn more about the differences between SOFRID and FORLI which lead to such fundamentally different averaging kernels and its implications when using the data.

*In the previous validation between SOFRID and FORLI with IAGOS (De Wachter et al., 2012), both algorithms had similar DFS and AKs. In the present study, SOFRID has been updated in order to perform simultaneous retrieval of CO and N2O as is discussed in the manuscript. The retrieval noise variance is reduced by a factor of 2 and the spectral window is enlarged, doubling the number of CO absorption lines and therefore enhancing the CO information content. Accordingly*

*the DFS is enhanced (about doubled) and the AKs are modified in SOFRID. No such changes were performed for FORLI. The implications of the differences in AK and DFS between both algorithms are discussed in the original manuscript in many occasions such as in:*

*- l238-242: "For SOFRID, the smoothing has little effect for the TCC (Table 2) and lower tropospheric columns (Table A2) but improves significantly the correlations for the mid-upper tropospheric columns (Table A3). For FORLI the variability ratios clearly decreaseand come closer to 1 and the Pearson's coefficients clearly increases for the 3 column".*

*- l269-272, in order to highlight better these implications, we have improved the discussion here. We discuss in more details the change in biases (and not only in absolute values of the biases as in the original manuscript) : "This is especially noticeable for FORLI* *at 12 out of 15 airports south of 29°N (latitude of Kuwait City) and at Shenyang where the biases are enhanced by more than 10% (up to 31%) when IAGOS data are smoothed as can be clearly seen in Fig. 6 (Table A2). Over Bogota, Caracas, Ho Chi Minh City, Dubai, Kuwait City, and Shenyang the biases are reduced in absolute value, resulting in a better agreement with IAGOS data when smoothing is applied. For SOFRID the biases are enhanced by 10 to 18% over Bogota, Lagos, Ho Chi Minh City, Hong Kong, Dubai and Shenyang resulting in an improved agreement with IAGOS data except at Hong Kong and Dubai (biases larger in absolute value)."*

*l279-280: "For FORLI, the application of the AKs brings the biases to large negative values south of Kuwait City..."*

*- l284-285 : "The lower impact of the AK smoothing on SOFRID comparisons result from the larger DFS for SOFRID retrievals (Table 1)".*

*See the detailed replies concerning DFS and AKs to reviewer #1 comments and corresponding manuscript modifications: l71 and l122.*

Smoothed IAGOS data is shown in Figures 6-12 as well as in Table 2 and in the appendix, but hardly ever discussed in the main text. I propose to either omit this discussion (which would also make the figures more readable) or to move respective figures and tables to the appendix. Instead, one could combine the IAGOS raw information from Tables 2, A2, and A3 into one table, replacing actual Table 2.

*The difference between raw and smoothed data is indeed hard to see in Figures 6-12 and we have removed the IAGOS smoothed data to make the figures more readable. We have consequently removed the few lines of text concerning smoothed data from the discussion of the timeseries. Nevertheless, the results concerning smoothed IAGOS data are kept in the corresponding tables (2, A2 and A3) and on Figure 6 because they are needed to assess the impact of the smoothing on the differences with the IAGOS data as discussed in section 3.3.1 and 3.3.2. The discussion improvement concerning the implication of AK smoothing on the biases (l269-272 see reply to previous comment) compensates for removing part of the text here. Table 2 concerning the TCC remains in the core of the manuscript because TCC is the most important and used product.*

Coming to my main point of criticism, the Figures are not carefully analysed and partly contain information which do not match the corresponding Tables and main text. I observed such clear mismatches in Figures 1, 5 and 6, which does not exactly strengthen my confidence in the veracity of the other figures.

*We tried to thorough fully analyze the figures and give as much details as possible about the different airports, layers and algorithms without too much confusion. We thank the reviewer for helping us to improve these analyses pointing to some mistakes and inadequacies.*

*We note that the reviewer was mistaken reading Figure 5. This figure provides a lot of information in a very synthetic way as is the aim of Taylor diagrams. We had a careful look at it that confirmed that our analysis and comments in the manuscript are correct (see replies to the specific points concerning l187 and 194-195 and also 207-208).*

*Otherwise, we have corrected the mistakes pointed to by the reviewer in the following comments.*

**Specific comments:**

Line 9: To be fair and also to align with the title, you should limit yourself to validation to the years 2008-2019, although a few 2020 validation data is available.

*We have removed the 2020 data from Frankfurt, the only airport concerned.*

Line 10: What is a "1.5 independent piece of information"? I think you should handle these as integers and talk of 1 to 3 independent pieces of information (also line 120).

*We do not agree. The DFS is not an integer and we keep 1.5 which means that we have more than one and less than 2 pieces of information which is not 1.*

Line 19-20 and several other occasions: A location specification such as "south of Bangkok" also implies spatial proximity to the location, which is not the case here. Please use latitude information to delineate the regions of interest.

*We modified : "south of 13.5°S (latitude of Bangkok)" or "north of 40°N (latitude of Philadelphia)"...*

Line 29: Please note that about 40-50% of methane sources are of natural origin.

*We added: "methane (CH4), which sources are up to 50% of natural origin"*

Line 44: The link to the pdf document should be transformed into a reference.

*Done*.

Line 47: I am missing an overview of the versions of IASI-CO retrieval and its potential effects on the validation. Are there any differences between IASI-A/B/C CO retrievals?

*See reply to general comment above and reply to reviewer #1 general comment.*

Line 50-52: I would not agree that anthropogenic pollution is generally the most important source of CO over Asia. You should rephrase the sentence to something like: "… covering a number of regions especially over Asia where anthropogenic pollution is enhanced …".

*Done*

Line 62: RTTOV needs to be explained.

*We added "Radiative Transfer for TOVS"*

Line 82: Please add Schlüssel et al. (2005) as reference to the IASI Level 2 Product Processing Facility.

*Done*

Line 84: You should explain what a "Level 1C error" stands for.

*Interferograms measured by IASI undergo three calibration levels to produce level 1C data (apodised spectra). At each calibration level, as at the moment of observation, an error can occur. line 84: "In addition, no retrieval is performed for pixels characterized by a Level 1C error due to instrument and/or processing, or by missing L2 EUMETSAT data."*

Lines 86-87: Are the general quality flag and the error profiles in any way taken into account for this study?

*Yes, we used the general quality flag. We add the following : "Only pixels with a general quality flag equal to 2 which correspond to the best quality are kept for the validation"*

Line 97: The study is about validation up to 2019. Please give more information on which 2020 data has been used and how it is included in the validation.

*The only data for 2020 were taken for Frankfurt by error. We have removed them. As they correspond to a very little part of the whole Frankfurt database, their removal do not change the results.*

Line 103: The Near and Middle East is usually counted as Asia. Please be more precise in the definition of the regions.

*We added the names of the airports: "Jeddah, Dubai, Doha and Kuwait City"*

Lines 105-106: Name the African airports here.

*Done: "Addis Ababa, Lagos, and Windhoek"*

Line 118: You rather mean here the validation time period or the validation days?

*We have changed "dataset" to "the whole validation period"*

Line 121-124: Describing the differences between SOFRID and FORLI algorithms are one of the major outcomes of the study. Elaborate more precisely on possible reasons which lead here to the fundamentally different profiles of the averaging kernels from the two algorithms.

*See reply to reviewer # 1 similar comment (l122) and modifications of the manuscript  and reply to your general comment about the same issue above. To summarize: in order to retrieve CO and N2O simultaneously, SOFRID has been updated with a twice larger spectral window almost doubling the number of CO absorption lines and the retrieval noise variance has been divided by 2. So the AKs and DFS have changed accordingly relative to De Wachter et al. (2012). FORLI did not undergo such modifications and its DFS and AKs are similar to those presented in De Wachter et al. (2012).*

Lines 126-128: Instead of speculating about grouping of the AKs one could simply give another figure displaying only the peak altitudes of the AKs as a function of their nominal height. Without this information I can't judge if two groups can clearly be distinguished for JJA.

*The AKs give theoretical and qualitative information about the retrieval that are clearly visible in our Figure 3: individual kernels, nominal altitude, kernels for the selected  columns. Given the very low vertical resolution of the retrievals (several kilometers), a high precision about the peak heights for individual kernels is not really meaningful. The partial columns of interest indeed encompass about half of the troposphere. We therefore prefer not to add another figure.*

Line 130: I'd rather see the lowest peak height at 800 hPa. Again, an additional figure as proposed before would help.

*We stated at "about" 900 hPa but we agree that it is rather at "800 hPa" and we have changed the number. This does not change the choice of the integrated partial columns.*

Line 187: TCC marker for New York in Figure 5 has approximately the same distance from the reference point in SOFRID and FORLI, resulting in a similar standard deviation.

*The referee has probably mistaken New-York (red squares) with another Airport (maybe Dallas with red circles) in Figure 5. For TCC, (right panels), the SOFRID (top panel) point for New-York is on the 1.0 circle with R=0.84 (see Table 2) and the FORLI one (bottom panel) is on a circle roughly corresponding to 1.2 with a R = 0.66 (see Table 2) and is therefore at a larger distance from the reference point [1.0, 0.0] resulting in a larger RMSD. We can therefore state that " the agreement is better for SOFRID at New-York".*

Line 194: I guess you wanted to give also the number of airports with R<0.5 but missed to do so.

*No, we wanted to write R² < 0.5 but the LaTeX code was not correct to write the square and just wrote R. We have corrected it.*

Lines 195-196: I cannot find confirmation of the R values for Düsseldorf in Fig. 5.

*We can clearly see Dusseldorf (light green diamonds) in Figure 5 for SOFRID (upper right panel) almost on the R=0.6 line between the circles centered at the origin [0.0;0.0] with radius 0.5 and 1.0. For FORLI (lower right panel), Dusseldorf appears almost at the intersection between the circle centered at the origin with radius 1.0 and the line corresponding to R = 0.3.*

Line 196: Is this always the case or just on average (as it can be deduced from Table 2)?

*The sentence l196 "on the contrary variabilities (standard deviations) are larger for FORLI than for SOFRID" implies that it is generally the case. This is supported by the details and numbers in the following part of the text l196-203.*

Lines 207-208: The only airport markers lying outside Fig. 5e are New York and Dallas.

*There is a misunderstanding: Dallas and NY are indeed outside of the 1.5 circle but are in Figure 5e. We state "For FORLI, Doha and Boston's variabilities are resp. 1.65 and 1.77 larger than IAGOS and the corresponding points are therefore out of the Taylor diagram". We give the values for these airports because they are not visible on the Figure. We have modified: "are therefore out of the Taylor diagram and not displayed on Figure 5e."*

LINE 251: Fig. 6 reveals median differences between -23% and 3%.

*Thank you for pointing this out. We have changed the numbers in the manuscript.*

Lines 255-256: Please use latitude information here instead of airport information.

*Done as everywhere necessary.*

Line 273: Fig. 6 reveals median differences between -20% and -1%.

*L273 statement is about mid-tropospheric (600-200 hPa) layer for SOFRID and FORLI (top panel of Figure 6). The lowest median bias is therefore -20% and the largest +8% (as stated in the manuscript). The largest bias for SOFRID only is -1%.*

Lines 281-284: This is an observation, not a conclusion. Of course, the reader would be interested in the conclusions from this finding. Use latitude bands!

*We think that the word "conclude" is correctly used here as "coming to the end of an analysis and summarizing it".  We have changed to latitude information.*

Line 289: Which datasets are you referring to? There are in total 5 datasets (SOFRID, FORLI, IAGOS raw and IAGOS smoothed in two ways, all for three atmospheric column types). All of these datasets are involved in Figures 7-12.

*We have changed to "the time series of the columns from the three datasets and of the differences between the IASI and IAGOS raw columns"*

Line 293: Can you confirm that summer biases are still not significant for time periods after 2012? In my opinion, this would require a statistical analysis.

The biases are systematically low rather than "not significant" in summer so we changed to : "and low biases in summer".

Lines 313-314: The seasonal and interannual bias variations are not as prominent than over Frankfurt due to the more incomplete temporal sampling.

*Modified.*

Lines 315-316: This can be omitted.

*We prefer to keep the statement.*

Line 355: 2020 -> 2019

*Done.*
* * *
Table A1: Add the number of profiles to each airport.

*Done. See below for the confusion in Figure 1 caption.*

Figure 1: The colour/size of the symbols used to characterize the airports do not agree with the numbers given in Table A1. E.g, three airports have more than 960 valid days as displayed in Figure 1, while in Table A1 only Frankfurt has more than 960 days with valid IAGOS profiles. In Table A1, 17 airports have less than 120 IAGOS days, while in Figure 1 only four airports have this characterization.

*The reviewer is right. There has been a confusion between the number of days and the number of profiles in the caption of Figure 1. At a given airport we can have multiple profiles the same day because of the same aircraft that take off and land and because more than one aircraft can visit an airport the same day (especially for big hubs like Frankfurt). In table A1 we provide the "number of days with valid profiles" which is therefore lower. The mistake is corrected with "days" replaced" by "profiles" in the caption of Figure 1. The text discussing Figure 1 was nevertheless correct with: "Frankfurt represents 35% (4917 profiles)…".*

Figure 2: What do the colours stand for? Ideally they should characterize latitude bands. Please explain and add an additional colour bar as legend.

*The colors are just there to better visually differentiate the airports. No need for a color bar.*

Figure 3: "validation database -> "validation days"?

*"Validation database at Frankfurt" means data selected for Frankfurt in the validation database.*

Figure 4: "AvKs" ->"AKs". IASI retrieval quality flag as well as MLS profile selection details and IAGOS data gap need to be explained (as part of the data section in the main text).

*MLS v5.0 CO profiles are selected according to recommendations in Livesey et al. (2020) (Status, Precison, Quality...) as mentioned in the text. We just report these recommendations in the flow chart for information. The reader should read Livesey et al. (2020) if interested in the details of MLS profile selection.*

*We added the following statement in the FORLI section for the quality flag in Fig. 4:* "Only pixels with a general quality flag equal to 2 which correspond to the best quality are kept for the validation."

*We added the following sentence in the SOFRID section to document what is understood by convergence in Fig. 4:* "We keep retrieved pixels for which convergence is achieved based on the value of the retrieval cost function (Jcost) output from the 1D-Var analysis which has to be positive. Jcost is positive if its fractional change between two consecutive iterations remains less than 0.01 (Haveman et al., 2020)"

*We have added the following text in the methodology section (where the selection of profiles and Figure 4 are presented) to explain the meaning of "data gap < 1500 m":* "Profiles must not show consecutive intervals of more than 1500 m in altitude without valid data".

Figures 5-12: You always start the discussion with the total column, followed by the partial columns. I'd therefore also group the figures in the same order (from left to right or from top to bottom, resp.). At least for Figures 6-12 it would be more intuitive to show 600-200 hPa in the middle and surface-600 hPa at the bottom. You should of course also alter the discussion order accordingly.

*We think that it is better to have the top altitude layer at the top of the plot, then the lower altitude layer below and then the sum of the two at the bottom for the best visual understanding of the plots. The order of discussion starts with the most used and popular product which is the TCC. We think it does not alter the understanding of the discussion.*

Figures 7-12: Timeseries for IAGOS raw and smoothed can mostly not be distinguished. Also IAGOS raw timeseries are partly covered by the smoothed timeseries. The only part of the text where you refer to the smoothed timeseries is lines 315/316. I propose that you remove the smoothed timeseries from the figures and also from the discussion.

*We agree that the plots are difficult to read with raw and smoothed data superposed. We therefore followed the reviewer recommendation to remove the smoothed data that are little discussed from the plots.* We modified and the discussion...

**Technical comments:**

Line 26: Please remove "still": *ok*

Line 28: Please add the year 2000 to the publication of Bergamaschi et al. : ok

Line 31: "makes of CO" -> "makes CO": *ok*

Line 34: "developped" -> "developed": *ok*

Line 36: add ")" at the end of the sentence: *ok*

Line 77: Please add a blank space before "cm": *ok*

Lines 94-95: "ascend and descend" -> "ascent and descent": *ok*

Line 95: "m.s$^{-1}$" -> "m s$^{-1}$": *ok*

Line 103: "important" -> "major" or "larger": *ok*

Line 113: AK has been explained before: *ok*

Line 181: "graduation" -> "marker"?: *Yes*

Line 273: "Instead of" -> "In contrast to": *yes*

Line 300: The same behaviour is observed: *Ok*

Line 305: "against" -> "compared to": *ok*

Line 360: "behaviours" -> "behaviour": *ok*

Line 382: For Taipei which is the airport with the second longest: *Ok*

Line 394: from May 14, 2019 onwards: *ok*

Line 425: Add the publication year (2000): *Ok*

**References:**

De Wachter, E., Barret, B., Le Flochmoen, E., Pavelin, E., Matricardi, M., Clerbaux, C., Hadji-Lazaro, J., George, M., Hurtmans, D., Coheur, P. F., Nedelec, P., and Cammas, J. P.: Retrieval of MetOp-A/IASI CO profiles and validation with MOZAIC data, Atmospheric Measurement Techniques, 5, 2843–2857, https://doi.org/10.5194/amt-5-2843-2012, 2012.

Schlüssel, P., Hultberg, T. H., Phillips, P. L. T., August, T., and Calbet, X.: The operational IASI Level 2 processor, Adv. Space Res., 36, 982, doi:10.1016/j.asr.2005.03.008, 2005.

*E. V. Stepanov, S.N. Kotelnikov, A.Y. Stavtsev, S.G. Kasoev, The best absorption lines for the detection of carbon monoxide at 2.35 micron with tunable diode lasers, Journal of Physics: Conference Series, https://dx.doi.org/10.1088/1742-6596/1560/1/012053, 1560, 2020.*

---

## Author Comment (AC2)

**Replies to RC1**: 'Comment on egusphere-2024-30', Anonymous Referee #1, 15 Apr 2024

We thank both reviewers for their thorough reading of the paper and their appropriate suggestions, which helped us improve our manuscript.

Black: reviewer comment
*Blue: authors reply*
*Red: manuscript correction*

General comment:

The other important source of validation data are the surface remote sensing FTIR instruments (NDACC). These are mentioned briefly but I would like to read more about the comparisons with the findings from these evaluations (e.g. EUMETSAT validation report). Are the main conclusions similar? Is the underestimate reported for the TCC quantitatively in agreement with FTIR evaluations?

*The main conclusions of the EUMETSAT validation report are : good agreement between FORLI-CO total columns from IASI/Metop-C and NDACC-FTIR data; average of the relative differences (IASI compared to NDACC stations) of 2.7%, average of the Pearson correlation coefficient of 0.89, (within values reported for IASI-A and -B). We have added these details to the text of the article.*
*Line 43: "According to a recent validation report (Langerock et al., 2021), FORLI-CO total columns from IASI/Metop-C show a very good agreement with NDACC-FTIR data with an average relative difference of 2.7% and a Pearson correlation coefficient of 0.89 . Furthermore this document shows that the distributions of IASI-A, -B and -C are highly consistent."*

Abstract:

l 5: It would be good to mention that MLS CO is used for the stratospheric part.

*CO is mostly a tropospheric compound and the most important are the IAGOS data. We therefore prefer not to mention MLS already in the abstract.*

l 9: Does the period "2008-2020" cover 12 or 13 years?

*The period is from beginning of 2008 to the end of 2019 : 12 years. We removed the only few data from 2020 at Frankfurt. This does not change the results for this airport due to the very large number of data from 2008 to 2019.*

l 12: What is meant by "to capture the CO variabilities"? Variability in time, or in the profile? I suggest to replace "variabilities" by "variability".

*Done*

l 13: Same question for "correlation coefficients". Does this refer to temporal correlations or vertical profile correlations?

*Correlations for the temporal variability. We added "for the timeseries".*

l 21: The last sentence "Our validation results will provide a better characterisation of IASI-CO data to the users and help improve the retrievals for future versions" can be formulated better. I would replace "will provide" by "do provide".

What does "better characterisation of IASI-CO data to the users" mean? Please reformulate "improve the retrievals for future versions".

*We have changed the statement to : "Our validation results provide an overview of the quality of IASI-CO retrievals to the users and insights for improving the retrievals in the future to the developers."*

l 36: Remove the "(" before "Hurtmans".

*Done*

l 47: The introduction gives the impression that the study of De Wachter 2012 is the only comparison with IAGOS. It would be good to mention that George 2015 also includes comparisons with IAGOS profiles, although this comparison is somewhat limited.

*De Wachter et al. (2012) is the only study to date to have performed a thorough validation with full timeseries of IAGOS profiles. We acknowledge that some comparisons were made in George et al. (2015): "George et al. (2015) also used some IAGOS profiles for comparisons with IASI-FORLI and MOPITT data."*

Introduction: The CO retrievals are introduced with a good set of references. But I am missing a paragraph in the introduction on IAGOS. What is it, what can be measured, some key achievements and key references, including validation work, to provide the reader with a background and further reading.

*We added a paragraph and references to better introduce IAGOS: "IAGOS uses commercial aircraft for automatic and routine in-situ measurements of atmospheric composition including reactive gases (e.g. ozone and CO), greenhouse gases, aerosols, and cloud particles along with essential thermodynamic parameters (Thouret al., 2006, Nedeec et al., 2015, Petzold et al., 2015). IAGOS provides regular observations in the upper troposphere and lower stratosphere (UTLS) during the cruise phase, and vertical profiles in the troposphere during landing and take-off, and in particular over regions that are never or poorly sampled. This long term quasi-global dataset has been used in a wide range of atmospheric studies, e.g. process studies, trend analysis, validation of climate and air quality models (Clark et al., 2021, Tsivlidou et al., 2023, Coheur et al. 2024), as well as for the calibration of space sensors and the validation of their retrievals (De Wachter et al., 2012, De laat et al. 2012)."*

Sec 2.1: The key reference for SOFRID is De Wachter. But this paper is more than 10 years old. Has there been any CO retrieval development in the meantime based on SOFRID? Are the updates made to SOFRID CO documented somewhere in more detail (is there a recent ATBD)?

*The product that is distributed until now is the one described in De Wachter et al. (2012) with minor changes. The major updates are described here to introduce the new version which includes simultaneous CO and N2O retrievals.*

l 70: "instead of operational EUMETSAT Level 2 IASI products". Why?

*The FORLI team collaborates with EUMETSAT and uses their L2 products for performing operational retrievals. The SOFRID team collaborates with colleagues from Météo-France/CERFACS (Toulouse) that are using ECMWF operational analyses in their assimilation system. To make comparisons between assimilated and retrieved O3 (Emili et al. 2019), in a coherent way it is better to use the same temperature and humidity products. The*

*SOFRID team therefore uses ECMWF operational analyses for SOFRID O3 and CO since a long time. As the results are satisfactory, they go on using ECMWF operational analyses.*

*The interesting point highlighted in the paper is that using ECMWF analyses with SOFRID allow to show that EUMETSAT Temperature discontinuities are probably responsible for temporal variations of CO columns biases in the FORLI data (see especially Figure 7, Frankfurt timeserie). Furthermore, some similar issues of continuity have been encountered in the FORLI-O3 data with a sudden drop in 2011 explained by changes in the EUMETSAT Temperature products (Boynard et al. 2016, 2018).*

*Nevertheless, this comment highlights that the phrasing with "instead" was misleading. We have therefore removed this statement.*

l 71: Could you please motivate why "The noise of the measurement covariance matrix has been reduced from 1.4 to $1.0 \cdot 10^{-8}$ W/(cm2 sr cm−1)" I assume that the performance of IASI has not improved over time?

*The retrieval noise is not IASI radiometric noise but needs to be tuned to take other error sources (errors in ancillary data such as temperature and humidity profiles, errors in RT modeling) and needs to be consitent with the Sa matrix. Here the retrieval algorithm is modified with a new version of the RT code, a larger spectral window, simultaneous retrieval of N2O and CO profiles, and the retrieval noise has to be re-tuned to re-optimize the retrievals. In De Wachter et al. (2012), the retrieval noise was chosen very conservatively in order to avoid providing anomalous CO values. In the version presented here, the experience and the use of long time series to test the retrievals allow to lower this parameter enhancing the information content and still avoiding retrieving noise. Furthermore, the N2O variability is much lower than the CO variability and it is important to tune the retrieval noise as low as possible to retrieve information about the N2O variability. We have improved the description as follows:*

*"The noise of the measurement covariance matrix has been reduced from 1.4 to $1.0 \cdot 10^{-8}$ W/(cm2 sr cm−1) in order to better capture the N2O variations. N2O spatio-temporal variations are indeed very low (less than 5%) compared to CO variations (one order of magnitude). This noise level is still very conservative and much larger than the radiometric noise of IASI-A estimated to be around $1.5 \times 10^{-9}$ W/(cm2 sr cm−1) in the CO spectral window (Clerbaux et al., 2009). The retrieval noise indeed takes other sources of errors into account such as errors on ancillary data (temperature and humidity profiles) or radiative transfer modeling errors. It was optimized with sensitivity tests performed on the CO IAGOS validation database."*

l 80: Same question as for SOFRID. The key ref for FORLI is from 2012 as well, again more that 10 years old. I noted the ATBD is from 2014. Are there updates compared to the 2012 Hurtmans paper? Any relevant evaluations of the FORLI CO retrieval published after 2012? Does FORLI v20151001 introduce any important changes compared to Hurtmans 2012?

*FORLI v20151001 is mainly a technical and spectroscopy update. We have added some details to the text of the article l87: "For this validation study FORLI-CO v20151001 was used. This version is an updated version from the one described in Hurtmans et al (2012), using look-up tables recalculated to cover a larger spectral range with a more recent version of the HITRAN spectroscopic database (HITRAN 2012) and implementing numerical corrections. It was validated with NDACC-FTIR data (Langerock et al., 2021). This version was installed..."*

l 97: What is the reason that "only airports providing at least 60 days with valid data " were selected? Is a number of 60 linked to the quality of the comparison? Even just one profile can still provide a useful comparison.

*We have used a threshold in order to have time series for each selected airport representative of CO temporal variations at their location. Airports with too limited number of profiles (some just provide a couple or a dozen of profiles) do not provide valuable information in our perspective: no time evolution and no seasonal or inter annual variability. Their use is rather counter productive in our general analysis aiming at characterizing the ability of the retrievals to capture the regional (latitudinal) and temporal variations of CO. We have added the following sentence at the end of the paragraph l111:*

*"The remaining airports provide temporally sparse profiles, which do not allow for sampling the temporal variabilities representative of their location."*

l 122: The difference in DFS is striking. The statement that "the reduction of the noise of the measurement covariance matrix relative to De Wachter et al. (2012)" is partly responsible askes for some more explanation. How can the noise be reduced by such a large factor? How does this compare to the a-priori noise assumed in FORLI? Please provide more detail.

*As mentioned in reply to comment l71 above, the retrieval noise has been reduced from 1.4 to 1.0 × $10^{-8}$ W (cm2 sr cm−1)−1 corresponding to a factor of 2 for the noise variance. It may appear to be important but the retrieval noise remains large compared to IASI estimated radiometric noise of 1.5x10-9 W (cm2 sr cm−1)−1. As mentioned in the manuscript, the retrieval window has also been extended from 2143–2181 cm−1 (De Wachter et al., 2012) to 2143-2218 cm−1 in order to improve N2O retrievals. The 0-1 CO absorption band is composed of its P branch below about 2140 and of the symmetrical R branch between 2140 cm-1 and 2225 cm-1. There are about as much CO absorption lines and information about CO in the 2181-2218 window than in the 2143-2181 cm-1 spectral range (Stepanov et al., 2020). The addition of the two effects leads to a significant increase of DFS documented in the paper. The DFS from FORLI and SOFRID were close to each other (between 1 and 2) in De Wachter et al. (2012) and there is indeed a "striking" difference now because of the changes in SOFRID detailed above. This was already explained in the manuscript: "The larger information content from SOFRID is due to (i) the extension of the spectral window and (ii) the reduction of the noise of the measurement covariance matrix relative to De Wachter et al. (2012). Both modifications are related to the combination of CO with N 2 O retrievals."*

*We have improved this explanation about the enhanced DFS in SOFRID with more details:*

*"In the former validation study (De Wachter, 2012), the SOFRID and FORLI DFS were close to each other, ranging between 1 and 2. The larger information content from SOFRID present version is due to two effects related to the simultaneous CO and N2O retrievals. First, the extension of the spectral window from 2143–2181 cm−1 (De Wachter et al., 2012) to 2143-2218 cm−1. The 2181-2218 cm−1 window indeed contains about half of the nu3 N2O absorption band (Barret et al., 2020). The 0-1 CO absorption band is composed of its P branch below about 2140 and of the symmetrical R branch between 2140 and 2225 cm-1 (Stepanov et al., 2020). The extension of the spectral window is therefore roughly doubling the number of CO absorption lines compared to De Wachter et al. (2012). Second, the retrieval noise variance has been reduced by a factor of 2 (see above) in order to improve the ability of the retrieval to capture N2O variations."*

l 119, 133: I note that TCC is used in two ways, either as total column (retrieval of 1 quantity) or as total atmosphere (for DFS = 2.9). Maybe better to use "Total atmosphere" instead of TCC in table 1.

*We have changed TCC for total atmosphere in Table 1 and in the text discussing DFS.*

l 177: I find the statement "because they provide the best assessment of the real differences between the in-situ and the remote sensed data" a bit dubious. Equation 1 describes how the retrieval relates to the real profile, and provides the best way of comparing. This is also evidenced by the validation results, e.g. l 238.

*Equation 1 allows to take the two largest sources of error into account for comparison: the effect of the a priori on possible biases and the smoothing of the true profile due to the limited vertical resolution. It is crucial for model validation because in that case the aim is to assess the quality of the model and not the quality of the satellite retrievals. Here, we want to assess the quality of the retrievals with reference in-situ data. The use of Equation 1 is indeed improving the comparisons but at the price of hiding the largest errors of the retrieved product and therefore prevents providing the complete evaluation of the product. A lot of papers only provide comparisons with smoothed profiles and the user cannot know the quality of the satellite product. We therefore consider the comparison with raw data as the "real" comparison because it allows a potential user to know what the product is worth but we also provide and discuss results using Equation 1.*

l 297: This mentions "two major updates of EUMETSAT Level 2 data processing". Could you please provide details on these updates in Sec. 2.

*These major updates of EUMETSAT Level 2 data processing improved retrieval of the vertical temperature profiles, and cloudy data flagging for the second one. We have completed with:* "These updates improved the retrieval of the vertical temperature profiles, and the cloudy data flagging for the second only."

l 377: "timeseries"

*Done*

l 387:What does "not statistically significant" refer to? From the paper I got the message that TCC globally is not really different between SOFRID/FORLI, but that negative biases are observed.

*The message is indeed close to what the reviewer reports. With significant, we meant that the RMSD was larger than the bias but this is not a sufficient condition to state that the bias is not significant so we removed "not statistically significant".*